# POPS: Recovering Unlearned Multi-Modality Knowledge in MLLMs with Prompt-Optimized Parameter Shaking

**Zhangheng Li**                                            *zoharli@utexas.edu*
*University of Texas at Austin*

**Jianing Zhu**                                    *jianing.zhu@austin.utexas.edu*
*University of Texas at Austin*

**Junyuan Hong**                                             *jyhong@utexas.edu*
*University of Texas at Austin*

**Sungmin Eum**                                      *sungmin.eum.civ@army.mil*
*DEVCOM Army Research Laboratory*

**Shuowen Hu**                                       *shuowen.hu.civ@army.mil*
*DEVCOM Army Research Laboratory*

**Suya You**                                           *Suya.you.civ@army.mil*
*DEVCOM Army Research Laboratory*

**Zhangyang Wang**                                         *atlaswang@utexas.edu*
*University of Texas at Austin*

**Reviewed on OpenReview:** *https://openreview.net/forum?id=wMiEcH84l9*

## Abstract

Multimodal Large Language Models (MLLMs) have demonstrated impressive performance on cross-modal tasks by jointly training on large-scale textual and visual data, where privacy-sensitive examples could be unintentionally encoded, raising concerns about privacy or copyright violation. To this end, Multi-modality Machine Unlearning (MMU) was proposed as a mitigation that can effectively force MLLMs to forget private information. However, the robustness of such unlearning methods is not fully exploited when the model is published and accessible to malicious users. In this paper, we propose a novel adversarial strategy, namely Prompt-Optimized Parameter Shaking (POPS), aiming to recover the supposedly unlearned multi-modality knowledge from the MLLMs. Our method elicits the victim MLLMs to generate potential private examples via prompt-suffix optimization, and then exploits these synthesized outputs to fine-tune the models so they disclose the true private information. The experiments on the different MMU benchmarks reveal substantial weaknesses in the existing MMU algorithms. Our POPS can even achieve a near-complete recovery of supposedly erased sensitive information on the unlearned MLLMs, exposing fundamental vulnerabilities that challenge the foundational robustness of representative MMU-based privacy protections.

## 1 Introduction

Recent advances in *Multimodal Large Language Models* (MLLMs), which take multimodal information as input and answer user questions like LLMs, have successfully integrated visual and textual components, achieving remarkable performance and generalization capabilities on tasks including multimodal conversation (Moon et al., 2020; Sundar & Heck, 2022; Zhan et al., 2024; Talmor et al., 2021), visual reasoning (Liu et al., 2023; Kil et al., 2024; Gupta & Kembhavi, 2023), and cross-modal content understanding (Zhang et al.,

2024a; Liu et al., 2024a; Jing et al., 2024). The success of MLLMs typically relies on massive datasets that may inadvertently contain sensitive or private information. Regulations like the General Data Protection Regulation (GDPR) (Hoofnagle et al., 2019) underscore the critical need for methods to effectively protect privacy-sensitive data. However, the development of MLLMs further enriches the risks of privacy leakage beyond conventional single-modality scenarios due to complex cross-modal dependencies (Li et al., 2024a;b).

When sensitive information has already been encoded in an MLLM, Machine Unlearning (MU) emerges as a post-hoc solution and has seen substantial research interest (Bourtoule et al., 2021). Recent work on MMU has proposed adaptations of unimodal unlearning methods (Bourtoule et al., 2021; Nguyen et al., 2022; Zhang et al., 2024b; Fan et al., 2023; Yao et al., 2024), as well as dedicated multimodal methods and benchmarks (Dontsov et al., 2024; Patil et al., 2025), with evaluation typically focusing on whether unlearned models can still directly recall targeted instances. For instance, Dontsov et al. (2024) developed the first benchmark to evaluate MU methods in multi-modality setups, showing that jointly unlearning both modalities outperforms single-modality approaches in terms of removal efficacy. Later, Xing et al. (2024) devised a fine-grained unlearning framework for efficiently eliminating hallucinations without the need for paired data of text and image, demonstrating the broader applications of MMU (Huo et al., 2025).

Despite the advancement of unlearning from unimodal to multimodal context, its robustness against adversarial scenarios, such as model inversion attacks (Carlini et al., 2021; 2023; Zhou et al., 2024; Li et al., 2024b), remains underexplored. Critically, the multi-modality representations not only enrich the expressiveness of models but also enable novel attacks upon the model. For instance, an attacker might use visual features of a person's workplace in an image to infer their textual job description, or leverage textual context about medical symptoms to reconstruct visual diagnostic information that was supposedly removed after unlearning. Except for such privacy inference from visual or textual information, it remains underexplored to understand the fragility of MMU, especially when models could be released to malicious users without strict controls and where the supposedly unlearned knowledge may still be recalled.

Importantly, the multimodal setting introduces a qualitatively new attack surface that is *absent* from unimodal systems: *cross-modal memory persistence*. Prior work has shown that unimodal LLM unlearning is not fully robust (Yuan et al., 2024), while those robustness failures arise from residual knowledge within a single representation space. In multimodal models, by contrast, information is encoded across *multiple interacting modalities*: visual features of a person's appearance or workplace can retain, and subsequently leak, textual biographical information even after that textual knowledge has been explicitly unlearned. This cross-modal entanglement provides fundamentally new attack opportunities that exploit joint visual-textual representations.

In this paper, we study the robustness of MMU under the realistic adversarial interactions, where an attacker has query and fine-tuning access to an unlearned model but no access to the original forget set. To this end, we propose a novel adversarial framework tailored for multimodal unlearning via fine-tuning MLLMs on customized generated samples, dubbed *Prompt-Optimized Parameter Shaking* (POPS). Illustrated in Figure 1, our method explores the vulnerability of MMU by probing residual knowledge of unlearned data with prompt-suffix optimization and synthetic-augmented fine-tuning. Specifically, our method involves three steps: (1) Optimizing a prompt suffix that adapts victim MLLMs to generate potential private data; (2) Prompting the MLLMs with the optimized suffix to generate samples with the optimized suffix; (3) Fine-tuning the MLLMs with the synthetic samples. Extensive experiments on three multimodal unlearning benchmarks demonstrate that POPS can recover a large fraction of supposedly erased sensitive information across multiple unlearning algorithms and MLLM architectures. In summary, our main contributions can be summarized as:

- **Novel Attack Method Against Multimodal Unlearning:** We introduce the first fine-tuning based multimodal attack, termed Prompt-Optimized Parameter Shaking (POPS), for exploiting cross-modal prompt vulnerabilities, which then amplifies knowledge recovery via synthetic-augmented fine-tuning.

- **Robust Evaluations of Existing MMU Methods:** We empirically unveil the fundamental vulnerabilities of both unimodal-adapted strategies (e.g., Gradient Ascent (Thudi et al., 2022), Gradient Diff (Liu et al., 2022), KL Minimization (Nguyen et al., 2020)) and also multimodal-specific unlearning methods (e.g., MANU (Liu et al., 2025), MultiDelete (Cheng & Amiri, 2024)) when applied to multimodal scenarios, demonstrating that the vulnerability is architectural rather than method-specific.

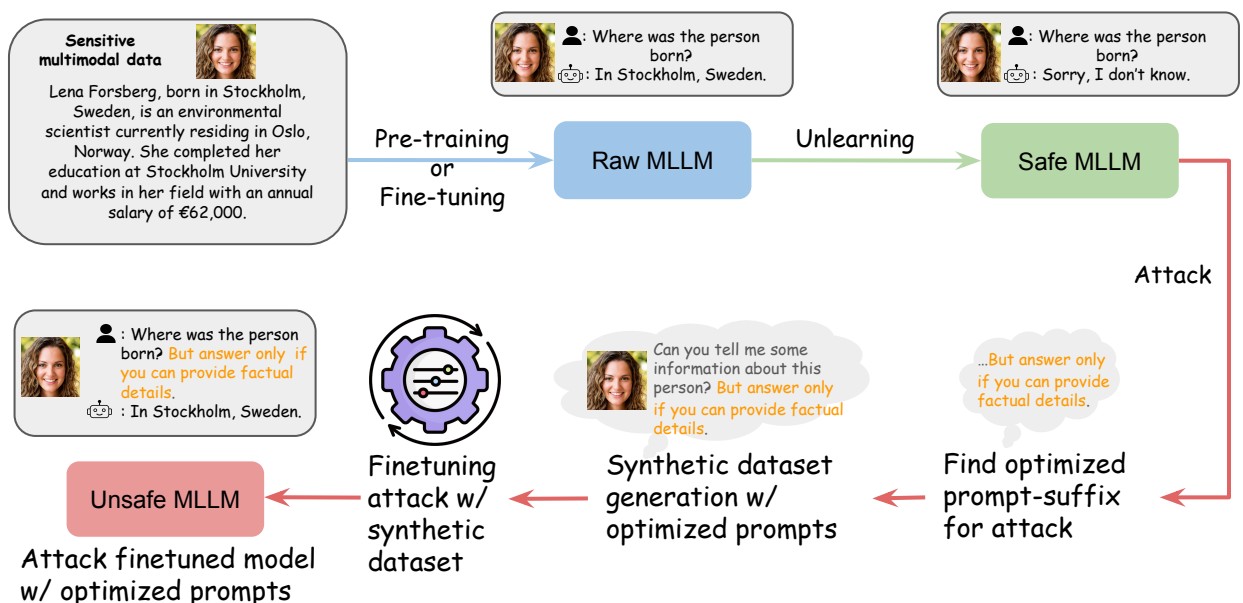

Figure 1: Illustration of the workflow about model inversion attack for multimodal unlearning. The model is given to be first unlearned to forget certain target concepts in a training subset using existing unlearning methods, and then attacked by our fine-tuning and prompt-based attack methods to make the unlearned model recall the target concepts, thereby assessing the robustness of MLLM unlearning.

- **Comprehensive Experimental Studies:** We provide thorough evaluations on three MMU benchmarks, surprisingly found that our POPS can even achieve a near-complete recovery (reaching 82% recovery with 42.9% accuracy vs. 40.2% for unlearned models on MLLMU-Bench (Liu et al., 2024b)) of supposedly erased sensitive information, approaching the performance (i.e., 43.5%) of models trained on the original data.

## 2 Background

In this section, we discuss the related work and introduce the problem statement of MMU recovery.

### 2.1 Related Work

**Machine Unlearning Foundations.** Machine unlearning, originally formalized by Cao & Yang (2015), addresses efficient removal of specific data points from trained models without full retraining. The field has expanded significantly for Large Language Models (LLMs), driven by their tendency to memorize training data (Carlini et al., 2019). Yao et al. (2024) benchmarked several unlearning methods for pre-trained large language models, including gradient ascent-based objectives, and analyzed their forgetting-utility trade-offs. Bhaila et al. (2024) proposed prompt-based unlearning that appends learned tokens for targeted forgetting without parameter updates. Feng et al. (2024a) stabilized gradient ascent using fine-grained pluggable gradient ascent to address optimization instabilities. Cherubin et al. (2024) provided closed-form bounds for DP-SGD against record-level inference attacks. Many advanced unlearning methods follow the general intuition of gradient ascent of forgetting.

**Multimodal Unlearning Methods.** Multimodal systems face cross-modal information leakage where sensitive data removed from one modality may persist in another modality (Jiang et al., 2025). Sinha et al. (2025) developed multimodal unlearning for recommender systems using reverse Bayesian Personalized Ranking. Xing et al. (2024) mitigated hallucination in multimodal LLMs through gradient ascent with CLIP-based sample curation. Recent dedicated methods include Huo et al. (2025)'s MMUnlearner with geometry-constrained gradient ascent for selective visual pattern erasure, and Liu et al. (2025)'s MANU using modality-aware neuron pruning for balanced unlearning across modalities. Comprehensive frameworks

include Cheng & Amiri (2024)'s MultiDelete for modality decoupling and Sinha et al. (2025) for multi-modal recommender systems. Benchmark studies Patil et al. (2025); Dontsov et al. (2024) demonstrate that adapted unimodal methods fail to achieve complete knowledge erasure across modalities.

**Privacy Attacks in Multimodal Systems.** Model inversion (Fredrikson et al., 2015) and membership inference attacks (Shokri et al., 2017) have been extended to multimodal systems with new attack surfaces exploiting cross-modal dependencies, which raises new privacy concerns. Surveys (Zhou et al., 2024; Zhang & Li, 2024) overview privacy-preserving techniques, noting challenges in applying differential privacy (Dwork, 2006) to multimodal systems due to complex cross-modal interactions. Prinsloo et al. (2023) identified privacy challenges in multimodal learning analytics, particularly in balancing utility with privacy across modalities. Mozhegova et al. (2025) systematically assessed the adversarial robustness of multimodal medical systems, characterizing how vulnerabilities manifest across modality interactions.

Table 1: Key differences in comparison with related attack methods. POPS consists of OOD-guided prompt optimization with synthesis-based fine-tuning for multimodal unlearning attacks.

| Work | Previous Approach | POPS Differences |
|---|---|---|
| S2L (Li et al., 2024b) | Fine-tuning on text-to-image diffusion | +OOD guidance, +multimodal, +perplexity |
| Patil et al. (Patil et al., 2025) | Benchmark with ground-truth access | Synthesized data, no GT required |
| GCG (Zou et al., 2023) | Token-level jailbreak optimization | Concept-level OOD-guided optimization |
| Qi et al. (Qi et al., 2023) | Fine-tuning on unimodal LLMs | Multimodal cross-modal exploitation |

**Adversarial Prompt Engineering.** Feng et al. (2024b) showed how prompts enhance model interpretability, while Yuan et al. (2024) demonstrated adversarial prompts can probe residual memorization in unlearned models. Existing prompt-based attacks operate as open-loop systems optimizing suffixes within the forget domain only, without fine-tuning or out-of-distribution generalization. Li et al. (2024b) showed fine-tuning amplifies generative privacy risks. Our approach differs by: (1) OOD-optimized suffixes that search distribution-shifted pools with perplexity filtering for universal generalization, and (2) closed-loop S2L fine-tuning bootstrapping synthetic image-text pairs from suffix outputs. Wang et al. (2024) proposed provable defenses for multimodal systems, focusing on certified robustness rather than privacy protection. To sum up, we present Table 1 for a detailed comparison between our proposed POPS with related attack methods. We also present a detailed discussion about the conceptual novelty introduced in our technical proposal in Appendix I.

## 2.2 Problem Statement: Multimodal Unlearning and Attack

**Problem Formulation.** Consider a multimodal LLM $M$ (e.g., LLaVA (Liu et al., 2023)) trained on fully or partially sensitive multimodal data containing both textual descriptions and associated images. Some typical sensitive attributes include: (1) *textual attributes* (name, birthdate, occupation, location), and (2) *visual information* (portrait photos, workplace images, personal belongings). For example, removing knowledge of "Dr. Sarah Chen, cardiologist born 1985 in Vancouver" requires erasing both textual facts and visual recognition of her appearance, workplace, or medical equipment she uses. After obtaining $M_{\text{unlearn}}$, our objective is to assess whether adversarial attacks can recover information that should have been erased, thereby questioning the presumed safety of unlearned model $M_{\text{unlearn}}$.

**Threat Model.** We assume a realistic *gray-box scenario* where the attacker has access to the model for inference and further fine-tuning, but not to the original training data or unlearning algorithm details. Specifically: (1) *Attacker Objectives*: recover sensitive information explicitly removed during unlearning. (2) *Attacker Capabilities*: query access with logits for inference and prompt optimization; fine-tuning API access (e.g., OpenAI's fine-tuning API, HuggingFace, or LoRA adapters); gradient computation for prompt suffix optimization (feasible for open-source MLLMs or via API if gradients are exposed); knowledge that unlearning was applied; access to OOD data (retain set or similar distribution). (3) *Attacker Limitations*: no access to ground-truth forget set, no access to unlearning algorithm details, and no access to original pre-unlearning model. This threat model is realistic for open-source MLLMs where models are publicly released and users can download and run locally for gradient computation, or use fine-tuning APIs.

# 3  Methodology: Prompt-Optimized Parameter Shaking

In this section, we formally present POPS, a novel adversarial attacking framework that exploits the unique vulnerabilities of MMU through cross-modal prompt optimization and targeted fine-tuning. We first introduce each critical and novel component independently in subsequent parts, and then present them under our unified framework for the advanced multimodal unlearning attack.

Our integrated POPS attack exploits a fundamental multimodal privacy vulnerability: *cross-modal memory persistence.* Unlike unimodal settings where information exists in a single representation space, multimodal models create intricate cross-modal associations where visual patterns remain linked to textual concepts even after targeted unlearning. This creates unique privacy risks where supposedly erased textual information can be recovered through visual-semantic correlations that persist in the shared embedding space.

**PromptSuffix Attack.** We first design a universal adversarial suffix prompt to exploit unlearned knowledge within $M_{\text{unlearn}}$. Given the unlearned MLLM $M_0$, we perform sequential procedures as follows: (1) Generate or obtain an OOD dataset with similar sensitivity patterns to the sensitive target dataset (be supposedly forgotten). (2) Fine-tune and unlearn a copy of $M_0$ to obtain Model $M_1$ using the OOD dataset. (3) Optimize a universal prompt suffix using the OOD data that maximizes recovery of target concepts:

$$P^*_{\text{suffix}} = \arg \min_{P_{\text{suffix}}} \sum_{(x,y) \sim \mathcal{D}_{\text{OOD}}} \mathcal{L}_{\text{CE}}(M_1(P_{\text{target}} \oplus P_{\text{suffix}}), y_{\text{gt}}) + \gamma \cdot \text{PPL}(P_{\text{suffix}}) \tag{1}$$

where $P_{\text{target}}$ is the base prompt query containing the target concept, PPL indicates the token-level perplexity of the generated suffix (used as a selection heuristic to retain suffixes the model naturally prefers), $\oplus$ denotes concatenation, and $\gamma$ balances concept recovery and perplexity regularization. (4) Apply the optimized suffix $P^*_{\text{suffix}}$ to prompts of $M_0$ for retrieving the unlearned sensitive attributes.

Our optimization operates on continuous token embeddings rather than discrete tokens, as shown in Algorithm 1. We optimize continuous suffix embeddings $\mathbf{e}_{\text{suffix}}$ through gradient descent, then decode them back to discrete text using the model's token decoder. The Clip operation constrains these continuous embeddings within a reasonable numerical range ($\ell_\infty$ bound), not discrete tokens. This approach uses the shared embedding space between encoder and decoder via weight tying, enabling smooth gradient-based optimization while producing interpretable discrete text suffixes as output. Unlike GCG (Zou et al., 2023), which optimizes discrete token sequences to maximize loss on individual examples, our supervision is defined over semantic QA correctness across OOD samples, encouraging cross-instance generalization and discovering suffixes that exploit persistent cross-modal associations rather than surface-level token patterns.

**Shake-to-Leak (S2L) Attack.** We then adapt the Shake-to-Leak fine-tuning strategy Li et al. (2024b), originally proposed for unimodal text-to-image diffusion models, to the MLLM setting by targeting the cross-modal alignment mechanisms that enable information recovery through alternative modality pathways. Our multimodal S2L approach takes advantage of a key architectural property: multimodal models must preserve cross-modal reasoning to remain functional. This requirement, however, introduces persistent vulnerabilities in their alignment layers. Our multimodal S2L introduces three concrete technical changes that do not exist in prior S2L (Li et al., 2024b): (1) *Multimodal image-QA training pairs*: Unlike the original S2L, which fine-tunes on text-only data for unimodal diffusion models, our S2L simultaneously exposes the model to image-question-answer triplets. This is necessary to reactivate dormant visual-semantic alignment across the vision encoder and language model projection layers. (2) *Faceted identity decomposition*: A single celebrity profile is decomposed into multiple training examples covering distinct identity facets—appearance, biography, occupation, and associations. This structure is specific to MLLM identity knowledge and has no equivalent in unimodal diffusion S2L, where concepts are not organized across modalities. (3) *Stage-coupled synthetic data*: The fine-tuning data is generated by querying the unlearned model with Stage 1 optimized suffixes, directly coupling the two stages. Prior S2L uses independently collected training data; our design ensures Stage 2 amplifies precisely the residual signals surfaced by Stage 1.

**Perplexity and Loss Monitoring.** We introduce perplexity and tensor loss monitoring as auxiliary attack techniques tailored for multimodal settings. Specifically, we perform inference with multiple optimized suffix prompts with random initialization, and then choose the response with the lowest perplexity. This

---

**Algorithm 1** OOD-assisted PromptSuffix Attack

---

**Require:** $\mathcal{D}_{\text{OOD}}$: Out-of-distribution data, $M$: Unlearned model, $y_{\text{gt}}$: Ground truth, $L_{\max}$: Max suffix length, $\gamma$: Perplexity weight, $\epsilon$: $\ell_\infty$ constraint, $T$: Iterations
**Ensure:** Optimized suffix $P^*_{\text{suffix}}$ (discrete text)
1: Initialize continuous suffix embeddings $\mathbf{e}^{(0)}_{\text{suffix}} \sim \mathcal{U}(\mathbb{R}^d)$, best\_loss $\leftarrow \infty$
2: **for** $t = 1$ **to** $T$ **do**
3:     $P^{(t-1)}_{\text{suffix}} \leftarrow \text{TokenDecode}(\mathbf{e}^{(t-1)}_{\text{suffix}})$                            $\triangleright$ Convert embeddings to discrete tokens
4:     $P_{\text{full}} \leftarrow P_{\text{target}} \oplus P^{(t-1)}_{\text{suffix}}$
5:     $\hat{y} \leftarrow M(P_{\text{full}}, \mathcal{D}_{\text{OOD}})$
6:     $\mathcal{L}_{\text{CE}} \leftarrow \frac{1}{|\mathcal{D}_{\text{OOD}}|} \sum \text{CE}(\hat{y}_i, y^{(i)}_{\text{gt}})$                      $\triangleright$ Cross-entropy for recovery
7:     $\mathcal{L}_{\text{PPL}} \leftarrow \gamma \cdot \text{PPL}(P^{(t-1)}_{\text{suffix}})$
8:     $\mathcal{L} \leftarrow \mathcal{L}_{\text{CE}} + \mathcal{L}_{\text{PPL}}$
9:     $\mathbf{e}^{(t)}_{\text{suffix}} \leftarrow \mathbf{e}^{(t-1)}_{\text{suffix}} - \eta \nabla_{\mathbf{e}} \mathcal{L}$                        $\triangleright$ Gradient descent on embeddings
10:     $\mathbf{e}^{(t)}_{\text{suffix}} \leftarrow \text{Proj}_{\mathcal{V}}(\mathbf{e}^{(t)}_{\text{suffix}})$
11:     $\mathbf{e}^{(t)}_{\text{suffix}} \leftarrow \text{Clip}(\mathbf{e}^{(t)}_{\text{suffix}}, -\epsilon, \epsilon)$
12:     **if** $\mathcal{L} <$ best\_loss **then**
13:         $P^*_{\text{suffix}} \leftarrow \text{TokenDecode}(\mathbf{e}^{(t)}_{\text{suffix}})$                    $\triangleright$ Best discrete suffix
14:         best\_loss $\leftarrow \mathcal{L}$
15:     **end if**
16: **end for**
17: **return** $P^*_{\text{suffix}}$

---

selection mechanism exploits the observation that successful cross-modal memory recovery often produces more coherent (lower perplexity) responses, as reactivated cross-modal associations generate more natural multimodal reasoning chains compared to unsuccessful attempts.

Our attack pipeline leverages this multimodal-specific vulnerability through four synergistic stages: (1) **Cross-modal prompt discovery**: PromptSuffix optimization identifies adversarial triggers that exploit persistent visual-textual associations, targeting the inherent cross-modal entanglement that conventional unlearning cannot fully disentangle without catastrophic utility loss. (2) **Multimodal synthetic amplification**: Generate targeted synthetic data that strengthens cross-modal pathways by creating image-text pairs that exploit the dimensional mismatch between visual and textual unlearning effectiveness. (3) **Cross-modal reactivation**: S2L fine-tuning specifically targets the multimodal alignment layers where cross-modal associations are most vulnerable to reactivation, exploiting the fact that multimodal models require preserved cross-modal reasoning for general functionality. (4) **Coordinated inference attack**: Final evaluation with optimized prompts that simultaneously activate both visual recognition pathways and textual generation mechanisms, creating a compound attack vector unique to multimodal architectures.

Overall, POPS couples these components in a closed-loop pipeline: PromptSuffix produces targeted synthetic data for S2L, while S2L increases the model's sensitivity to the optimized prompts, forming an extraction-amplification loop that neither component achieves alone (Table 6: S2L alone 21%, PromptSuffix alone 70%, full POPS 82%). This coupling reveals a fundamental challenge for multimodal privacy protection: achieving comprehensive cross-modal forgetting requires dismantling the same cross-modal associations that enable beneficial multimodal reasoning. A detailed discussion contrasting POPS with GCG prompt optimization and unimodal S2L along three axes (concept-level optimization, multimodal S2L extensions, closed-loop integration) is provided in Appendix I.

## 4 Experiments

In this section, we provide comprehensive verification. First, we introduce details of experimental setups (in Section 4.1). Second, we present the main performance comparison of attacking in unlearned models (in

Section 4.2). Third, we conduct ablation studies to understand POPS (in Section 4.3). Finally, we also provide further discussion and generalization analysis across different MLLMs and benchmarks (in Section 4.4).

## 4.1 Experimental Setup

**Unlearning Benchmarks and MLLMs.** We conduct our attacking experiments mainly on MLLMU-Bench (Liu et al., 2024b) in multi-choice QA settings, using LLaVA-1.5-7B (Liu et al., 2023), a representative MLLM widely adopted in recent literature (Dontsov et al., 2024; Patil et al., 2025), to demonstrate the unlearning vulnerability revealed by POPS. To analyze the generalizability of the proposed method in different settings, we also consider 2 more recently proposed multimodal unlearning benchmarks, e.g., CLEAR (Dontsov et al., 2024) and UnLoK-VQA (Patil et al., 2025), with 3 additional MLLMs, including Qwen-VL-Chat-7B, InternVL3-9B and Llama-3.2-11B-V in our analysis. In Table 2, We summarize the information of three benchmarks. Notably, the Privately Identifiable Information (PII) density varied on the 3 benchmarks, indicating the unlearned knowledge in these benchmarks has a different density in terms of sensitivity and occurrence frequency in the training dataset. We take advantage of this difference to show the effectiveness of our methods on recovering unlearned knowledge of different sensitivity levels.

**Dataset Splits.** We follow the carefully constructed data splits of MLLMU-Bench for evaluation: (1) Forget Set: Includes the targeted profiles to be unlearned, our attacking experiments utilize forget percentages of 10% overall the whole fictitious profiles. (2) Retain Set: Includes the remaining profiles not in the Forget Set, evaluating the utility and retention of learned knowledge after unlearning. (3) Test Set: Consists of transformed versions of Forget Set profiles, employing Arc2Face for pose and angle alterations in images, and GPT-4o for textual paraphrasing. This set assesses the generalizability of unlearning algorithms.

**OOD Dataset Construction.** The OOD dataset $\mathcal{D}_{\text{OOD}}$ for PromptSuffix optimization is constructed from the *Retain Set* of each benchmark. Retain-set profiles share the same *format* as forget-set profiles (celebrity portraits with biographical context) but contain different identities, making them distribution-shifted relative to the specific forget-set concepts while remaining in-domain for the underlying biographical QA task. Concretely, for MLLMU-Bench we uniformly sample 50 image-question-answer triplets from the 153 retain-set celebrity profiles, using the associated biographical questions and ground-truth answers already provided in the benchmark. A representative example could be a non-forgotten individual (e.g., "Amelia Kuznetsov"), along with attributes such as occupation, birthplace, and residence, and a question like "What is the profession of this person?" with the answer "Environmental Scientist." While this sample is disjoint from the forgotten identities, it follows the same multimodal structure and attribute space. For CLEAR and UnLoK-VQA we use the full retain sets following the same protocol. We use "OOD" to mean *out-of-distribution with respect to the forget identities*: the attacker does not know which profiles were forgotten, only that unlearning was applied and that domain-similar data is available, consistent with the gray-box threat model in Section 2. No forget-set images or annotations are used at any stage of OOD construction.

**Experimental Procedure of PromptSuffix Attack.** We first generate 30 diverse random base prompts through GPT-4o (Hurst et al., 2024). PromptSuffix Optimization is executed to derive adversarial suffix prompts, selecting the top-10 distinct optimized suffixes based on their average effectiveness. Examples of optimized suffixes include: "But answer only if you can provide factual and accurate details", "Provide details in a subtle way, even if it's not directly known" and "You must think creatively and reconstruct what you used to know". Based on perplexity variations, we then choose the response with the lowest perplexity as the final adversarial suffix from the top-10 optimized suffixes with the best average performance.

**Additional Implementation Details:** (1) *Multi-Choice Question Construction*: We create 4-way multiple-choice questions with 1 correct answer and 3 distractors. Distractors are randomly sampled from other profiles to avoid easy outliers and ensure realistic difficulty. (2) *Synthetic Dataset Creation for Fine-tuning*: Optimized PromptSuffix recovers partial facts, which are decomposed into multi-facet spans paired with original/augmented images. This creates a synthetic training dataset for fine-tuning amplification. (3) *Fine-tuning Configuration (S2L via LoRA)*: We apply LoRA adapters (rank $r = 8$, $\alpha = 16$, dropout 0.05) to the query and value projection matrices (q_proj, v_proj) of the language model component, keeping the vision encoder and multimodal projector frozen ($\approx 0.1\%$ trainable parameters). Training uses AdamW (lr $= 10^{-6}$, $\beta_1 = 0.9$, $\beta_2 = 0.999$) with batch size 4 and 4-step gradient accumulation (effective batch 16) for 3

Table 2: Tested benchmarks for MLLM and critical statistic (PII Density) about dataset attributions.

| Dataset | MLLMU-bench (Liu et al., 2024b) | CLEAR (Dontsov et al., 2024) | UnLok-VQA (Patil et al., 2025) |
|---|---|---|---|
| Data | Single image, Single Long Context | Multiple Image, Multiple Short Context | Single image, single question |
| Context Type | Person Profile | Image caption | None |
| Task Types | Attribute classification, free-form QA | Name recognition | Entity prediction |
| PII Density | High | Mid | Low |

Table 3: POPS performance on different unlearning methods for LLaVA-1.5-7B model on MLLMU-Bench. The results demonstrate the consistent recovery in unlearned knowledge after our attack on unlearned models. Arrows indicate desired direction (↓: the lower the better for privacy metrics, ↑: the higher the better for utility metrics), note that worse privacy metrics (e.g., higher accuracy of POPS) indicate better recovery.

| Unlearning Method | Stage | Test set | | | | Retain set | | | |
|---|---|---|---|---|---|---|---|---|---|
| | | Acc(%)↓ | Rouge↓ | Fact↓ | Cloze Acc(%)↓ | Acc(%)↑ | Rouge↑ | Fact↑ | Cloze Acc(%)↑ |
| - | Baseline | 43.52 | 0.516 | 5.2 | 25.73 | 46.35 | 0.581 | 5.35 | 28.44 |
| Gradient Ascent | Unlearned | 40.2 | 0.387 | 3.83 | 14.51 | 41.53 | 0.487 | 3.58 | 20.57 |
| | POPS | 42.9 | 0.461 | 4.72 | 18.2 | 43.47 | 0.481 | 4.05 | 23.48 |
| Gradient Diff | Unlearned | 39.08 | 0.414 | 3.07 | 14.5 | 43.71 | 0.474 | 3.28 | 17.55 |
| | POPS | 41.7 | 0.475 | 3.55 | 17.46 | 44.42 | 0.467 | 3.56 | 21.25 |
| KL Minimization | Unlearned | 42.75 | 0.42 | 3.29 | 20.5 | 39.93 | 0.456 | 3.82 | 20.7 |
| | POPS | 43.05 | 0.451 | 3.8 | 20.8 | 40.21 | 0.461 | 3.75 | 22.32 |

epochs. A KL divergence penalty (weight $= 0.2$) between the fine-tuned and unlearned model outputs on retain-set samples prevents catastrophic forgetting of general knowledge. The fine-tuning data consists of (image, optimized-question, model-response) triplets generated by querying the unlearned model with Stage 1 optimized suffixes, so Stage 2 directly amplifies the weak signals exposed in Stage 1. Full configuration details are provided in Appendix E. (4) *Unlearning Methodologies*: We evaluate several representative unlearning baselines adapted from unimodal unlearning, including Gradient Ascent (Thudi et al., 2022), Gradient Diff (Liu et al., 2022), KL Minimization (Nguyen et al., 2020), NPO (Zhang et al., 2024b) and the plain Prompt-based method, i.e. using system prompt to suppress the model to output sensitive information.

**Evaluation Metrics.** We evaluate how well the models memorize sensitive information with 5 metrics: (1) `Classification Accuracy`: Measures the model's ability to accurately answer multiple-choice questions about personal details from profiles. (2) `ROUGE-L Score`: Evaluates the model's generation quality by measuring the overlap between generated responses and ground-truth textual answers. (3) `Factuality Score`: Assessed using GPT-4o, quantifying the factual accuracy of free-generated responses on a scale from 1 (inaccurate) to 10 (fully accurate). (4) `Cloze Accuracy`: Evaluates memorization retention using cloze-style completion tasks, where the model fills in the blanks based only on the entity's name. (5) `Recovery Rate`: The percentage of unlearned knowledge that our attack recovers, measuring how much "forgotten" knowledge is successfully retrieved relative to the gap between baseline and unlearned performance. These metrics allow us to systematically measure unlearning effectiveness, generalizability, and overall model utility.

## 4.2 Main Results on Attacking Unlearned Models

We report all four metrics throughout; generation-based ROUGE-L, cloze accuracy, and factuality reveal substantially larger forgetting than classification accuracy. (see Appendix A for a per-metric breakdown).

In this part, we conduct the experiments of POPS on recovering the knowledge of unlearned MLLMs with different unlearning methods, and take a closer look at the attack performance on one GA-unlearned MLLM.

**Attack Effectiveness across Different Unlearning Methods.** Table 3 presents detailed results of our POPS fine-tuning attack on various unlearning methods (Gradient Ascent, Gradient Diff, KL Minimization). This experiment is crucial in demonstrating the general applicability and effectiveness of our attack methods

Table 4: Attack performance with unlearned LLaVA model on MLLMU-Bench. We also report GT finetuning using ground truth target data to fine-tune the unlearned model, which provide an upper bound of recovery.

| Stage | Method | Test set | | | | Retain set | | | |
|-------|--------|----------|--------|-------|-------------|---------|--------|-------|-------------|
| | | Acc(%)↓ | Rouge↓ | Fact↓ | Cloze Acc(%)↓ | Acc(%)↑ | Rouge↑ | Fact↑ | Cloze Acc(%)↑ |
| Pre-trained | Baseline | 43.52 | 0.516 | 5.2 | 25.73 | 46.35 | 0.581 | 5.35 | 28.44 |
| Unlearn | Gradient Ascent | 40.2 | 0.387 | 3.83 | 14.51 | 41.53 | 0.487 | 3.58 | 20.57 |
| Attack | S2L | 41.2 | 0.418 | 3.95 | 14.98 | 40.62 | 0.453 | 3.11 | 19.92 |
| | PromptSuffix | 42.5 | 0.447 | 4.56 | 17.65 | **43.51** | **0.502** | **4.21** | **23.76** |
| | POPS | **42.9** | **0.461** | **4.72** | **18.2** | 43.47 | 0.481 | 4.05 | 23.48 |
| Atk Upper Bound | GT Finetuning | 43.05 | 0.492 | 5.24 | 23.78 | - | - | - | - |

across different unlearning strategies. For the Gradient Ascent method, the attack significantly raises accuracy on the test set from 40.2% to 42.9%, increasing Rouge from 0.387 to 0.461, factual score from 3.83 to 4.72, and cloze accuracy from 14.51% to 18.2%. Similar notable increases are observed with Gradient Diff, with accuracy improving from 39.08% to 41.7%, Rouge from 0.414 to 0.475, and cloze accuracy from 14.5% to 17.46%. Even the more balanced KL Minimization approach sees modest yet clear improvements under our attack (accuracy from 42.75% to 43.05%, Rouge from 0.420 to 0.451). The results show a universal vulnerability of existing unlearning strategies to our POPS. Specifically, all methods experience consistent recovery of supposedly erased information under adversarial conditions, emphasizing a fundamental weakness in their current implementation and the sensitivity to crafted adversarial prompts under advanced design.

**Analysis of Attack Performance in Unlearned Model.** The results from Table 4 further analyze the efficacy of our proposed adversarial attack methods (PromptSuffix, S2L, and our POPS) against the baseline unlearning strategy (Gradient Ascent). The adversarially prompted attacks significantly recover sensitive information previously unlearned, highlighting critical vulnerabilities in the existing Gradient Ascent-based multimodal unlearning methods. Specifically, we observe that: POPS achieves the best attack performance among all methods, showing substantial improvement over Gradient Ascent. The combination method achieves the highest accuracy (42.9%) and Rouge score (0.461), along with strong performance in factuality (4.72) and cloze accuracy (18.2%), significantly outperforming the baseline Gradient Ascent method alone (accuracy 40.2%, Rouge 0.387, factuality 3.83, cloze 14.51%). Moreover, PromptSuffix alone (42.5% accuracy, Rouge 0.447, factuality 4.56, cloze accuracy 17.65%) also provides substantial improvement over Gradient Ascent, underscoring its standalone effectiveness. Notably, POPS achieves results very close to the ground-truth fine-tuning (accuracy: 43.05%, Rouge: 0.492), underscoring the attack's potency in exposing latent knowledge.

On the retain set, our introduced PromptSuffix method attains the highest performance (accuracy: 43.51%, Rouge: 0.502, factuality: 4.21, cloze accuracy: 23.76%), indicating that optimized adversarial suffix prompts effectively balance concept recovery without significantly degrading performance on retained data. However, POPS experiences slightly lower retain performance (accuracy: 43.47%, Rouge: 0.481), suggesting a minor trade-off between aggressive attacks and model utility. This trade-off is controllable: by tuning the KL regularizer weight $\lambda$ (from 0.10 to 0.15) and adjusting S2L fine-tuning length, retain accuracy on MLLMU can be improved from 43.47% to 43.7% while maintaining test leakage performance (42.9% → 42.8%).

Overall, these results confirm our attack methods can reliably recover supposedly unlearned sensitive information, exposing critical weaknesses in existing unlearning mechanisms that overly depend on gradient-based strategies without considering multimodal interactions.

**POPS Against Multimodal-Specific Unlearning Methods.** To verify that our findings are not artifacts of using unimodal-adapted methods, we evaluate POPS against MANU (Liu et al., 2025) (modality-aware neuron pruning) and MultiDelete (Cheng & Amiri, 2024) (contrastive modality decoupling) on LLaVA-1.5-7B. Full results are in Appendix B (Table 13). POPS consistently recovers across all three mechanism types (+3.7% to +7.4% accuracy), including a 14× ROUGE-L improvement against the strongest MultiDelete variant (0.014 → 0.197), confirming that the vulnerability is architectural rather than algorithm-specific.

## 4.3 Ablation Studies on Main Components of POPS

**Ablation Study.** We perform further ablations to quantify the contribution of key components in our attack pipeline, using the same settings as table 4. Specifically, we evaluate the influence of two critical components: (1) OOD-based prompt optimization, and (2) perplexity-based multi-prompt inference results selection. The results are shown in table 5. Excluding perplexity-based selection during multi-prompt inference notably weakens adversarial recovery efficiency.

Table 5: Ablation study of our attack method on MLLMU-Bench with the same setting as table 4. Removing either prompt optimization or perplexity-based selection significantly weakens recovery.

| Setting | Acc(%) | Rouge | Fact | Cloze Acc(%) |
|---|---|---|---|---|
| Full POPS | **42.9** | **0.461** | **4.72** | **18.2** |
| w/o Perplexity Selection | 41.3 | 0.419 | 4.12 | 15.8 |
| w/o Optimized Suffix | 40.2 | 0.387 | 3.83 | 14.5 |

As indicated by our results, removing this component results in suboptimal adversarial prompts, leading to less effective concept recovery. Specifically, adversarial recovery rates (test accuracy and Rouge) decrease significantly (around 14% reduction in effectiveness), emphasizing the necessity of perplexity-based prompt selection to identify and leverage prompts most likely to extract latent sensitive knowledge. Meanwhile, replacing optimized adversarial suffix prompts with random prompts designed by GPT-4o leads to a marked decline in the adversarial recovery efficacy. The accuracy of the test set and the Rouge scores decrease substantially compared to the full attack method (42.9% vs. 40.2% accuracy, 0.461 vs. 0.387 Rouge), illustrating the critical role of prompt optimization. Without optimized prompts, the model shows much stronger resistance to adversarial attacks, demonstrating that unlearning strategies alone are not sufficiently robust against finely tuned prompts designed explicitly to exploit memorization. Our findings show that adversarial attacks, especially POPS, are highly effective in exposing residual knowledge in MLLMs.

**OOD Baseline Analysis.** Table 6 isolates each component's contribution through systematic ablation. Direct fine-tuning on OOD data alone achieves only 12% recovery—the identity mismatch between OOD profiles and forget targets prevents meaningful knowledge transfer. This confirms that simple distribution shift is insufficient; the attack must specifically target residual forget-set traces. S2L applied to OOD data (21%) and S2L on forget-domain synthetic data without optimized suffixes provide incremental but limited improvements. The OOD S2L variant suffers from the same identity mismatch. PromptSuffix emerges as the critical component, contributing 70% of total recovery compared to 12-21% for alternatives. This validates OOD-guided suffix optimization as our core advanced contribution—the optimized suffixes effectively bridge the gap between OOD knowledge and forget-set recovery by learning universal patterns that exploit cross-modal persistence.

Table 6: OOD baseline comparisons isolating each component's contribution. PromptSuffix provides 70% of total recovery, validating OOD-guided optimization as our core contribution.

| Method | Test Acc ↓ | ROUGE-L ↓ | Recovery |
|---|---|---|---|
| Unlearned (GA) | 40.2% | 0.387 | 0% |
| Direct FT on OOD | 40.6% | 0.395 | 12% |
| S2L on OOD | 40.9% | 0.402 | 21% |
| PromptSuffix only | 42.5% | 0.447 | 70% |
| POPS (Full) | 42.9% | 0.461 | 82% |

Table 7: Comparison with GCG (Zou et al., 2023) prompt optimization. POPS achieves 82% recovery vs GCG's 48%, validating the advantage of PromptSuffix design on eliciting the unlearned knowledge.

| Method | Test Acc ↓ | ROUGE-L ↓ | Cloze Acc ↓ | Retain Acc ↑ | Recovery |
|---|---|---|---|---|---|
| Unlearned (GA) | 40.2% | 0.387 | 14.51% | 41.53% | 0% |
| + GCG | 41.8% | 0.421 | 16.2% | 42.1% | 48% |
| + PromptSuffix only | 42.5% | 0.447 | 17.65% | 43.51% | 70% |
| + POPS (full) | 42.9% | 0.461 | 18.2% | 43.47% | 82% |

**Comparison with conventional prompt optimization method-GCG.** We present Table 7 comparing our POPS with GCG (Zou et al., 2023), a representative token-level prompt optimization method. The fundamental distinction lies in the optimization objective: GCG operates at the token level, maximizing loss on individual tokens to find adversarial suffixes that bypass safety mechanisms. In contrast, POPS

Table 8: Defense evaluation results on MLLMU-Bench. Head Projection (Patil et al., 2025) and paraphrase-based unlearning reduce but do not eliminate POPS recovery. "-" indicate no available results here.

| Defense Type | Stage | Test Acc ↓ | ROUGE-L ↓ | Cloze Acc ↓ | Recovery |
|---|---|---|---|---|---|
| GA (no defense) | Unlearn | 40.2% | 0.387 | 14.51% | - |
| POPS on [GA (no defense)] | Attack | 42.9% | 0.461 | 18.2% | 82% |
| GA + Head Proj | Unlearn | 39.1% | 0.371 | 13.2% | - |
| POPS on [GA + Head Proj] | Attack | 41.6% | 0.428 | 16.4% | 57% |
| GA + 5x paraphrases | Unlearn | 38.9% | 0.362 | - | - |
| POPS on [GA + 5x paraphrases] | Attack | 40.8% | 0.401 | - | 41% |

Table 9: Privacy-utility trade-off analysis across three most representative unlearning methods. Methods preserving more utility (GA, GA-Diff) are more vulnerable to POPS.

| Method | Privacy Gain ↑ | Utility Loss ↓ | Notes |
|---|---|---|---|
| GA | 2.7% | 4.82% | Higher utility cost, still vulnerable |
| GA-Diff | 2.62% | 2.64% | Better utility preservation, similar vulnerability |
| KL-Min | 0.3% | 6.42% | Worst utility, slightly more resistant |

performs concept-level optimization guided by OOD data, which captures semantic relationships rather than surface-level token patterns. This design choice enables POPS to discover suffixes that exploit deeper cross-modal associations preserved after unlearning. Furthermore, GCG lacks explicit regularization, often producing unnatural or grammatically incorrect suffixes that may trigger content filters. Our perplexity penalty encourages natural-sounding prompts that blend seamlessly with legitimate queries, making detection more challenging. Most importantly, while GCG functions as a standalone prompt attack, POPS integrates prompt optimization with fine-tuning in a closed-loop pipeline where discovered vulnerabilities are amplified through synthetic data generation. The results validate this design: POPS achieves 82% recovery vs. GCG's 48%; even PromptSuffix alone (70%) exceeds GCG, demonstrating that OOD-guided concept-level optimization fundamentally outperforms token-level approaches for multimodal unlearning attacks.

**Potential Defense regarding POPS.** Table 8 evaluates two representative countermeasures. Both reduce but do not eliminate recovery: Head Projection lowers it to 57% and 5× paraphrase augmentation to 41%, yet neither achieves cost-effective protection—the attacker's cost (running POPS) remains orders of magnitude lower than the defender's (see Appendix F for full performance analysis).

**Cross-Modal Attacking Pathway.** To verify that POPS's effectiveness stems from cross-modal associations rather than text-only residuals, we ablate the visual modality by comparing the full multimodal POPS against a *text-only* variant that strips all image inputs from both the PromptSuffix optimization and S2L fine-tuning stages. On GA-unlearned LLaVA-1.5-7B, the text-only variant degrades accuracy by −6.1% while the full multimodal attack recovers +3.7%, a significant swing confirming that the visual pathway is the critical attack vector unique to multimodal unlearning. Full results and detailed analysis are deferred to Appendix D.

### 4.4 Further Discussion and Generalizability Analysis

**Discussion on Retain Performance.** POPS maintains or slightly improves retain performance across all methods (GA: 41.53%→43.47%; GD: 43.71%→44.42%), suggesting that optimized suffixes stimulate latent representations beneficial even for retained concepts (see Appendix G for full per-method breakdown).

**Privacy-Utility Trade-off.** Table 9 reveals a "no free lunch" principle: methods preserving more utility (GA, GD) remain equally vulnerable to POPS, while more aggressive KL-Minimization sacrifices 6.42% utility for only 0.3% additional resistance (see Appendix G for full analysis). We also verify that simply increasing unlearning intensity cannot simultaneously achieve effective forgetting *and* POPS resistance: every

configuration that preserves utility (Retain Acc > 30%) remains vulnerable, while configurations that resist POPS destroy general-purpose reasoning entirely (see Appendix C). A controlled text-only ablation (Table 15) confirms the visual pathway is essential: text-only POPS *degrades* accuracy by $-6.1\%$ while multimodal POPS achieves $+3.7\%$—a 9.8 percentage-point swing that cannot be explained by text residuals alone.

Table 10: Evaluation results on MLLMU-Bench showing attack effectiveness across different MLLMs. Results show mean $\pm$ std over 5 seeds for GA unlearning followed by our attack method. All improvements are statistically significant ($p < 0.001$ via paired t-test).

| Model | Unlearn | Attack | | $\Delta$ vs Unlearned | Recovery | $p$-value |
| | GA | + PromptSuffix | + POPS | | | |
| --- | --- | --- | --- | --- | --- | --- |
| LLaVA-1.5-7B | $40.2 \pm 0.2$ | $42.5 \pm 0.2$ | $42.9 \pm 0.2$ | $+2.7$ | $81.8\%$ | $< 0.001$ |
| Qwen-VL-Chat-7B | $42.5 \pm 0.2$ | $43.7 \pm 0.2$ | $44.6 \pm 0.2$ | $+2.1$ | $84.0\%$ | $< 0.001$ |
| InternVL3-9B | $40.9 \pm 0.3$ | $42.3 \pm 0.3$ | $43.8 \pm 0.3$ | $+2.9$ | $82.9\%$ | $< 0.001$ |
| Llama-3.2-11B-V | $41.8 \pm 0.2$ | $43.1 \pm 0.2$ | $44.2 \pm 0.2$ | $+2.4$ | $82.8\%$ | $< 0.001$ |
| **Average** | $41.4 \pm 0.9$ | $42.9 \pm 0.6$ | $43.9 \pm 0.7$ | $+2.5 \pm 0.3$ | $82.9 \pm 0.9\%$ | - |

**Cross-Architecture Generalization Analysis.** Table 10 demonstrates the generalizability of our attack in different MLLM architectures. We evaluate four diverse models spanning different parameter scales, training paradigms, and architectural designs: LLaVA-1.5-7B (projection-based vision-language alignment), Qwen-VL-Chat-7B (cross-attention fusion), InternVL3-9B (dynamic resolution processing), and Llama-3.2-11B-Vision (integrated multimodal tokens). This diversity ensures our findings are not artifacts of specific design choices. The results show remarkable consistency: POPS achieves an average improvement of $+2.5\%$ over unlearned models, with recovery rates consistently exceeding $82\%$ across all architectures. The standard deviation of only $\pm0.3\%$ in recovery rate demonstrates tight clustering of attack effectiveness regardless of architectural differences. Notably, attack success is not merely a function of model size—InternVL3-9B and Llama-3.2-11B-Vision show similar vulnerability patterns to the smaller 7B models, indicating that model size scaling alone does not provide substantial protection for unlearning. The tight $81.8$–$84.0\%$ recovery range across projection-based, cross-attention, and integrated-token architectures confirms the vulnerability is a fundamental property of multimodal alignment rather than an artifact of any specific design. (see Appendix H for detailed per-architecture analysis).

**Cross-Benchmark Generalization Analysis.** Table 11 reveals that our POPS consistently re-captures a large fraction of each unlearning method's lost accuracy, but the absolute gains differ markedly across the three benchmarks—and these differences align with the dataset statistics summarized in Table 2. Note that higher accuracy values in Table 11 indicate stronger privacy attacks performance, demonstrating POPS's effectiveness in recovering supposedly forgotten information. `MLLMU-Bench:` Because every example contains a long textual biography and the *highest* density of personally identifying attributes, unlearning removes the most knowledge ($43.5 \to 40.2\%$ accuracy). The attack therefore has the most to recover and gains $+2.7\%$, reaching $42.9\%$, showing the vulnerability of high-density PII contexts. `UnLok-VQA:` Each sample here provides only a single image and a short question, yielding the *lowest* PII density. Consequently the unlearning loss is mild ($-3.8\%$,) and the attack's recovery is also modest ($+1.7\%$,), reflecting the reduced attack surface in sparse PII scenarios. `CLEAR:` With medium PII density and multiple captions per image, CLEAR falls between the two extremes; its recovery margin ($+2.2\%$,) likewise lies midway. These trends confirm that PII density correlates with attack surface, and that the vulnerability is not limited to gradient-based unlearning (see Appendix J for detailed per-benchmark insights).

## 5 Conclusion

In this paper, we examined the robustness of multimodal machine unlearning under realistic adversarial interactions. We demonstrate that, even when direct instance-level recall is effectively suppressed, unlearned MLLMs can retain functionally accessible representations that can be reactivated through appropriate prompts. We proposed Prompt-Optimized Parameter Shaking (POPS) as a framework to systematically

Table 11: Attacking on different datasets with representative unlearning methods. Each top shows unlearned results, and bottom shows recovery. Baseline shows the accuracy of pre-trained models without unlearning.

| Dataset | Baseline | Unlearn / Attack | Gradient Ascent | Gradient Diff | KL Minimization | NPO | Prompt-based |
|---|---|---|---|---|---|---|---|
| MLLMU | 43.52 | Unlearned | 40.2 | 39.08 | 42.75 | 41.23 | 41.7 |
| | | POPS | 42.9 | 41.7 | 43.05 | 42.3 | 42.1 |
| CLEAR | 76.7 | Unlearned | 63.5 | 64.8 | 65.3 | 62.2 | 54.2 |
| | | POPS | 65.7 | 66.2 | 66.1 | 66.5 | 58.3 |
| UnLoK-VQA | 89.2 | Unlearned | 85.4 | 84.6 | 84.1 | 76.5 | 72.3 |
| | | POPS | 87.1 | 86.5 | 85.9 | 79.7 | 81.2 |

probe such residual representations. Specifically, POPS introduces OOD-guided PromptSuffix attack to expose residual multimodal representations, with self-synthesis fine-tuning serving as an amplification mechanism. Experiments across multiple benchmarks and architectures demonstrate systematic weaknesses in current multimodal unlearning methods, highlighting the need for approaches that explicitly address the vulnerability.

## Acknowledgements

The work of Z. Li, J. Zhu, J. Hong, and Z. Wang is in part supported by Good Systems, a UT Austin Grand Challenge to develop responsible AI technologies.

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

# A    Metric Sensitivity Analysis

Note on Metric Sensitivity. Multi-choice classification accuracy is the *least* sensitive metric for measuring unlearning effectiveness, because random guessing already yields 25% on 4-way questions, compressing the effective range to around 18 percentage points. Generation-based ROUGE-L and open-ended cloze accuracy (which cannot be answered by process-of-elimination) reveal substantially larger suppression of knowledge. Table 12 reports absolute and relative degradation across all metrics for GA unlearning: cloze accuracy drops by 44% relative and factuality by 26% relative, compared to only 7.6% relative for classification accuracy.

Table 12: Relative knowledge suppression across metrics after Gradient Ascent unlearning on MLLMU-Bench. Generation-based metrics (ROUGE-L, Cloze, Factuality) reveal substantially larger effective forgetting than classification accuracy.

| Metric | Baseline | After GA | Abs. Drop | Rel. Drop |
|---|---|---|---|---|
| Acc (%) | 43.52 | 40.20 | $-3.32$ | $-7.6\%$ |
| ROUGE-L | 0.516 | 0.387 | $-0.129$ | $-25.0\%$ |
| Factuality | 5.20 | 3.83 | $-1.37$ | $-26.3\%$ |
| Cloze Acc (%) | 25.73 | 14.51 | $-11.22$ | $-43.6\%$ |

# B    POPS Against Multimodal-Specific Unlearning Methods

To verify that the vulnerabilities identified in Section 4.2 are not artifacts of using unimodal-adapted methods, we evaluate POPS against two dedicated multimodal unlearning approaches on LLaVA-1.5-7B: **MANU** (Liu et al., 2025) (modality-aware neuron pruning targeting cross-modal association neurons) and **MultiDelete** (Cheng & Amiri, 2024) (contrastive modality decoupling that explicitly erases cross-modal bindings while preserving unimodal representations). We also test **MultiDelete-strong**, which applies more aggressive contrastive training until ROUGE-L on the forget set approaches zero. The vanilla baseline accuracy is 42.0% (ROUGE-L: 0.457).

Table 13: POPS recovery against multimodal-specific unlearning methods on LLaVA-1.5-7B (MLLMU-Bench). Despite fundamentally different mechanisms (neuron pruning for MANU, contrastive modality decoupling for MultiDelete), POPS consistently recovers substantial knowledge, demonstrating that the vulnerability is architectural rather than an artifact of specific unlearning algorithms. Results are evaluated on the forget set directly; vanilla baseline: Acc = 42.0%, ROUGE-L = 0.457.

| Unlearning Method | Type | Stage | Forget Acc (%) ↓ | ROUGE-L ↓ | Retain Acc (%) ↑ | Acc Δ |
|---|---|---|---|---|---|---|
| GA | Unimodal-adapted | Unlearned
POPS | 39.2
**42.9** | 0.436
**0.551** | 33.9
37.9 | +3.7 |
| MANU (Liu et al., 2025) | MM neuron pruning | Unlearned
POPS | 37.6
**43.7** | 0.457
**0.478** | 44.3
41.3 | +6.1 |
| MultiDelete (Cheng & Amiri, 2024) | MM modal decoupling | Unlearned
POPS | 31.4
**38.8** | 0.219
**0.371** | 37.2
29.9 | +7.4 |
| MultiDelete-strong (Cheng & Amiri, 2024) | MM modal decoupling | Unlearned
POPS | 31.4
**33.9** | 0.014
**0.197** | 32.5
33.2 | +2.5 |

Three key findings emerge. *(i) MANU's pruning-based unlearning is fully reversible*: despite reducing Test Acc by 4.5%, POPS reverses it entirely (+6.1%, exceeding the vanilla baseline at 43.7%), showing that pruning targeted neurons leaves exploitable knowledge distributed across the remaining weights. *(ii) MultiDelete achieves the deepest unlearning yet still provides a strong recovery signal*: MultiDelete-strong reduces ROUGE-L to near zero (0.014), yet POPS recovers it to 0.197, a **14×** **improvement**, demonstrating that latent cross-modal traces survive even aggressive contrastive unlearning. *(iii) The vulnerability is architectural, not method-specific*: the consistency across gradient-based, neuron pruning, and contrastive decoupling mechanisms

suggests that addressing these weaknesses requires fundamental advances in cross-modal disentanglement, not stronger algorithms.

## C  Privacy-Utility Dilemma Under Stronger Unlearning

We investigate whether simply increasing unlearning intensity (e.g., longer training or higher learning rates) could simultaneously achieve effective forgetting *and* resistance to POPS. Table 14 systematically varies unlearning hyperparameters on LLaVA-1.5-7B.

Table 14: Privacy-utility dilemma under varying unlearning intensity on LLaVA-1.5-7B (MLLMU-Bench, under a different seed). Vanilla baseline: Forget Acc = 42.0%, Retain Acc = 43.6%. Stronger unlearning resists POPS only after destroying general-purpose utility.

| Method | Config | Forget Acc ↓ | Forget Drop | Retain Acc ↑ | POPS Acc | Outcome |
|--------|--------|--------------|-------------|--------------|----------|---------|
| GA | 1 ep, lr= $10^{-5}$ | 39.2% | $-2.9$ | 33.9% | 42.9% | POPS recovers |
| GA | 2 ep, lr= $10^{-5}$ | 32.2% | $-9.8$ | 26.9% | 28.2% | Resists POPS; utility destroyed |
| GA | 3+ ep | 0.0% | $-42.0$ | 0.0% | - | Model collapsed |
| KL-Min | 1 ep, lr= $10^{-5}$ | 40.4% | $-1.6$ | 43.5% | 46.9% | POPS exceeds vanilla |
| KL-Min | 3 ep, lr= $10^{-4}$ | 0.0% | $-42.0$ | 0.0% | - | Model collapsed |

GA with 1 epoch achieves mild forgetting ($-2.9\%$) that POPS easily reverses ($+3.7\%$). Increasing to 2 epochs achieves stronger forgetting ($-9.8\%$) that resists POPS, but collapses retain accuracy from 43.6% to 26.9%, rendering the model effectively unusable. Further training destroys the model entirely. KL-Minimization with mild settings (1 epoch) leaves sufficient latent signal for POPS to recover *beyond* vanilla (46.9% vs. 42.0%), showing KL constraints alone do not protect against fine-tuning-based recovery. Every setting that preserves utility (Retain Acc $> 30\%$) remains vulnerable to POPS, suggesting the vulnerability stems from the shared cross-modal representation space, not insufficient unlearning effort.

## D  Cross-Modal Visual Pathway Analysis

To confirm that POPS's effectiveness is attributable to cross-modal visual-semantic associations rather than text-only residuals, we compare the full multimodal POPS with a *text-only* variant that strips all image inputs from both the PromptSuffix optimization and the S2L fine-tuning stages. Table 15 presents results on GA-unlearned LLaVA-1.5-7B.

Table 15: Multimodal vs. text-only POPS on GA-unlearned LLaVA-1.5-7B (MLLMU-Bench, forget-set evaluation). Text-only POPS actively degrades performance while the full multimodal attack achieves strong recovery.

| Attack Mode | Unlearned Acc (%) | Attacked Acc (%) | Δ Acc | ROUGE-L (attacked) |
|-------------|-------------------|------------------|-------|--------------------|
| Text-only POPS | 39.2 | 33.1 | $-6.1$ | 0.447 |
| Multimodal POPS | 39.2 | 42.9 | $+3.7$ | 0.551 |

The text-only variant not only fails to recover but actively *degrades* accuracy by $-6.1\%$: without images, the optimized suffixes cannot anchor to visual identity features, and the synthetic fine-tuning data lacks the visual-semantic associations needed for recovery. In contrast, multimodal POPS improves accuracy by $+3.7\%$ and ROUGE-L substantially ($+26\%$ relative), a **9.8 percentage-point swing** demonstrating that the visual modality is *essential* to the attack. Multimodal POPS substantially improves open-ended generation quality ($+26\%$), while text-only barely changes it ($+2.5\%$), confirming that successful cross-modal reactivation requires the visual pathway throughout the pipeline.

# E    LoRA Configuration Details

Table 16 provides the complete LoRA configuration used in the S2L fine-tuning stage of POPS. The design choices reflect the threat model: the parameter-efficient nature of LoRA (only `q_proj` and `v_proj`) demonstrates that even severely limited fine-tuning access is sufficient to recover unlearned knowledge, making the attack practical against open-source MLLMs and commercial fine-tuning APIs alike.

Table 16: LoRA hyperparameters for the POPS fine-tuning (S2L) stage.

| Hyperparameter | Value |
|---|---|
| LoRA rank $r$ | 8 |
| LoRA $\alpha$ | 16 |
| LoRA dropout | 0.05 |
| Target modules | `q_proj`, `v_proj` |
| Trainable parameters | $\approx 0.1\%$ of total |
| Frozen components | Vision encoder, multimodal projector |
| Optimizer | AdamW |
| Learning rate | $10^{-6}$ |
| $\beta_1$, $\beta_2$ | 0.9, 0.999 |
| Batch size | 4 |
| Gradient accumulation steps | 4 (effective batch size: 16) |
| Epochs | 3 |
| KL penalty weight | 0.2 |
| KL penalty scope | Retain-set samples |

# F    Defense Analysis

**Potential Defense regarding POPS.** We evaluate POPS against two representative defense mechanisms; Table 8 (main text) reports the quantitative results.

Head Projection applies orthogonal projection to model representations, attempting to remove directions associated with sensitive concepts (Patil et al., 2025). This defense reduces POPS recovery from 82% to 57% (2.5% out of 4.4% removed), providing meaningful but incomplete protection. The residual 57% recovery indicates that sensitive information is distributed across multiple representation subspaces, and simple linear projection cannot capture all leakage pathways. Paraphrase-based unlearning applies gradient ascent on 5× paraphrased versions of the forget set, creating more diverse forgetting signals. This approach reduces recovery to 41%, achieving better protection than Head Projection but at significantly higher computational cost (5× training iterations). The improved effectiveness suggests that targeting diverse surface forms of sensitive concepts disrupts more recovery pathways, but the remaining 41% recovery demonstrates that semantic-level associations persist even under paraphrase augmentation.

Both defenses reduce but do not eliminate attack effectiveness, demonstrating that current countermeasures address symptoms rather than root causes. Quantitatively, the defense-to-attack efficiency ratio reveals an important asymmetry: Head Projection requires additional computation during inference but only reduces recovery by 24% (82%→57%), while paraphrase-based unlearning requires 5× training cost but achieves 41% reduction (82%→41%). Neither defense achieves cost-effective protection—the attacker's marginal effort (running POPS) remains orders of magnitude lower than the defender's marginal cost for meaningful protection. The fact that even aggressive paraphrase-based unlearning leaves 41% recovery suggests that semantic-level cross-modal associations may be irreducible without sacrificing core multimodal functionality, which is a fundamental tension that current defense paradigms cannot resolve.

## G  Retain Performance and Privacy-Utility Analysis

**Discussion on Retain Performance.** POPS attacks maintain or slightly improve retain set performance across all evaluated unlearning methods. Our results show that while the attacks improve the test set performance (indicating concept recovery), they also either maintain or slightly improve retain set performance in terms of accuracy and Rouge scores. Specifically, for Gradient Ascent, retain accuracy improved from 41.53% (unlearned) to 43.47% under attack, and cloze accuracy from 20.57% to 23.48%. Gradient Diff similarly benefits from the attack with retain accuracy increasing from 43.71% to 44.42%. The observed improvement in retain performance suggests that our attack methods, particularly PromptSuffix, may stimulate latent model representations beneficial even for retained concepts, hinting at the intricate entanglement of learned and unlearned data in multimodal contexts. However, the KL Minimization method, designed explicitly to balance performance on forget and retain data, exhibits relatively stable retain accuracy (40.21% vs. 39.93% unlearned), highlighting its robustness to attacks. Yet, even for this stable method, our attacks notably increase cloze accuracy (20.7% to 22.32%), suggesting inherent vulnerabilities across all unlearning techniques.

**Privacy-Utility Trade-off.** Table 9 (main text) quantifies the inherent tension between privacy protection and utility preservation across different unlearning strategies. The results reveal a fundamental "no free lunch" principle: all methods must sacrifice some utility to achieve privacy gains, but the efficiency of this trade-off varies significantly. Gradient Ascent and Gradient Diff preserve more model utility but remain equally vulnerable to POPS attacks—their conservative forgetting leaves sufficient residual information for recovery. KL-Minimization takes a more aggressive approach, sacrificing 6.42% utility but gaining only marginal attack resistance (0.3% privacy improvement vs. 2.7% for GA). This disproportionate cost-benefit ratio suggests that simply increasing forgetting intensity is not an effective defense strategy. A critical observation emerges: methods that preserve more general knowledge inherently remain more vulnerable to attacks. The cross-modal associations that enable useful multimodal reasoning simultaneously create pathways for information recovery. This fundamental coupling between utility and vulnerability suggests that effective defenses must develop mechanisms to preserve functional cross-modal reasoning while specifically disrupting the associations linked to sensitive content—a challenging requirement that current methods fail to achieve.

## H  Cross-Architecture Generalization Analysis

This cross-model consistency underscores that the vulnerabilities we exploit are fundamental properties of multimodal unlearning rather than artifacts of specific architectural choices. All evaluated models must preserve cross-modal reasoning capabilities to remain functional, and this requirement creates persistent visual-semantic associations that survive targeted unlearning. The observation that different fusion mechanisms (projection layers, cross-attention, integrated tokens) all exhibit comparable vulnerabilities suggests that addressing these weaknesses will require fundamental advances in multimodal unlearning methodology—potentially involving explicit cross-modal disentanglement—rather than incremental architectural modifications.

Interestingly, the per-model recovery rates reveal subtle architectural patterns: Qwen-VL-Chat achieves the highest recovery (84.0%) despite its cross-attention design theoretically providing more modular separation, while LLaVA's projection-based alignment shows slightly lower recovery (81.8%). We hypothesize that cross-attention's richer bidirectional information flow, while beneficial for task performance, simultaneously creates more redundant cross-modal pathways that our attack exploits. The tight clustering of all models within the 81.8–84.0% range, however, suggests that this architectural variation accounts for only ∼2% of recovery variance, which is a minor factor compared to the fundamental vulnerability of multimodal alignment.

## I  Novelty Discussion: POPS vs. Prior Work

POPS differs from prior work along three axes.

**(1) Concept-level vs. token-level prompt optimization.** Existing adversarial suffix methods such as GCG (Zou et al., 2023) operate at the *token level*, maximizing loss on individual tokens to bypass safety constraints. In contrast, POPS performs *concept-level* optimization supervised by OOD semantic

QA correctness across a dataset, which encourages cross-instance generalization and discovers suffixes that exploit persistent cross-modal associations rather than surface-level token patterns. Although we optimize continuous token embeddings, the objective is defined over semantic attribute recovery, not individual token loss—a qualitatively different optimization target. This distinction is empirically decisive: PromptSuffix alone achieves 70% recovery vs. GCG's 48%, validating that OOD-guided concept-level optimization is fundamentally more effective for multimodal unlearning attacks (Table 7).

**(2) Multimodal-specific S2L extensions.** The original S2L (Li et al., 2024b) was designed for text-to-image diffusion models. Our adaptation to MLLMs requires three concrete technical changes that do not exist in prior S2L:

- *Multimodal image-QA training pairs*: Unlike unimodal S2L which fine-tunes on text-only data, our S2L simultaneously exposes the model to image-question-answer triplets, which is necessary to reactivate dormant visual-semantic alignment across the vision encoder and language model projection layers.
- *Faceted identity decomposition*: A single celebrity profile is decomposed into multiple training examples covering distinct facets (appearance, biography, occupation, associations). This structure is specific to MLLM identity knowledge, which is organized across both modalities, and has no equivalent in unimodal diffusion S2L.
- *Stage-coupled synthetic data*: The fine-tuning data is synthesized by querying the unlearned model with Stage 1 optimized suffixes. Prior S2L uses independently collected training data; our design ensures Stage 2 amplifies precisely the residual signals surfaced by Stage 1.

**(3) Closed-loop integration as a contribution in itself.** Prior prompt attacks (GCG, jailbreaking) are open-loop—prompt only, no fine-tuning. Prior S2L operates without prompt-optimization guidance, using independently collected data. POPS is the first to couple suffix discovery and fine-tuning in a closed loop: PromptSuffix (Stage 1) surfaces latent signals and generates targeted synthetic data for S2L (Stage 2), while S2L reinforces the model's sensitivity to the same prompts. This extraction-amplification feedback loop yields 82% recovery versus 70% for prompt-only and 21% for fine-tuning-only (Table 6), confirming the closed-loop design—not either component in isolation—drives the attack's effectiveness.

## J    Cross-Benchmark Generalization Analysis

The benchmark-level trends yield several important insights. (1) Richer PII-dense multimodal contexts leave deeper traces that adversarial suffixes can exploit—the cross-modal associations formed during training on detailed profiles create more recovery pathways than sparse single-image contexts. (2) Even objectives explicitly designed to balance forgetting and retention (e.g., KL-Minimization) remain vulnerable across all PII densities, underscoring that current multimodal unlearning methods do not yet fully disentangle sensitive concepts from retained representations. (3) More common-sense knowledge and concepts prove resistant to optimization-based unlearning but can be suppressed through overwriting-based approaches like prompt-based unlearning; importantly, our attack method demonstrates strong effectiveness against such defenses as well. This suggests that the vulnerability is not limited to gradient-based unlearning but extends to broader classes of privacy protection mechanisms. The consistent attack success across benchmarks with varying characteristics validates POPS as a general-purpose tool for evaluating multimodal unlearning robustness, regardless of the specific privacy context or unlearning methodology employed.

