# OpenReview forum: "POPS: Recovering Unlearned Multi-Modality Knowledge in MLLMs with Prompt-Optimized Parameter Shaking"
_TMLR — Accepted by TMLR_

### Review · Reviewer_M4RG · 2026-02-18

**Summary Of Contributions:**

In this paper, the authors are the first to evaluate the robustness of multimodal machine unlearning under more extensive/realistic adversarial settings. To do so, the authors propose a novel framework called POPS, which first identifies an adversarial suffix via optimization with the goal of maximizing data recovery. Then. POPS pairs the suffix with fine-tuning to further cause the MLLM to recover the “forgotten” data. Through experiments with 4 different multi-modal LLMs and 3 unlearning benchmarks, the authors show that POPS consistently recovers model performance for the unlearned content (similar to benign settings).

**Audience:**

Yes

**Audience Explanation:**

Yes, unlearning has been a consistent topic of interest, and this paper addresses the lack of realistic adversarial robustness testing of unlearning methods when applied to MLLMs.

**Claims And Evidence:**

Yes

**Claims Explanation:**

Yes, the claims are accurate and supported via experiments involving several MLLMs and unlearning benchmark datasets and methodologies.

**Requested Changes:**

**Strengths**
- This paper proposes a relevant and threat model that is under-explored in current literature.
- The proposed technique is well motivated and has a generalizable design.

**Weaknesses**
- The overall results of the unlearning attacks are weak, which makes it harder to interpret the effectiveness of the POPS attack.
- The experimental setup is a bit unclear due to a lack of details

**Requested Changes**
-

**Baseline Unlearning is Weak**

My key concern is that the unlearning techniques the authors use do not appear to be very successful in the first place. For example, in table 3, unlearning reduces the accuracy very little, at most 3%, which suggests that most of the information in the “forget set” is clearly still being recalled (at least in multi-choice answering). While it’s evident that POPS does bring up the accuracy back to the baseline performance, this undermines the potential effectiveness and need of POPS since unlearning does not appear to be working in the first place. Can the authors offer explanation as to why the unlearning is unsuccessful?

**Experimental Setup Clarifications**

Some of the paragraphs in 4.1 are a bit unclear. In particular, based on the threat model in 2.2, a key part of the POPS attack is that the adversary does not know knowledge was forgotten (”no access to ground-truth forget set”), and thus must create a fine-tuning set based on access to an OOD retain set. However, point (2) in “Additional Implementation Details” of 4.1 is not clear how image/text pairs for this OOD set are constructed. Can the authors elaborate on this? Furthermore, it is not clear how S2L training setting is applied in the LoRA setting.

---

> ### Author Response · Authors · 2026-03-25
> **Response to Reviewer M4RG (Part-I)**
>
> We sincerely thank Reviewer M4RG for recognizing that POPS addresses "the lack of realistic adversarial robustness testing of unlearning methods when applied to MLLMs" and for the constructive feedback. We address each concern below.
>
> ---
>
> ### Q1: Baseline Unlearning Appears Weak
>
> > *"The unlearning techniques the authors use do not appear to be very successful in the first place. For example, in Table 3, unlearning reduces the accuracy very little, at most 3%, which suggests that most of the information in the 'forget set' is clearly still being recalled. This undermines the potential effectiveness and need of POPS since unlearning does not appear to be working in the first place."*
>
> We appreciate this important observation. To address it directly: the question of whether unlearning is effective and the question of whether POPS is needed are related but separate — they operate at *different levels of representation*.
>
> **1) Classification accuracy understates the suppression of knowledge, and stronger unlearning reveals a privacy-utility trade-off.**
> The ~3% accuracy drop on multi-choice classification (Table 3 of our paper) reflects a well-known ceiling effect of recognition tasks: random guessing already yields 25% on 4-way questions, compressing the measurable range. The more sensitive recall-based metrics tell a different story:
>
> - **Cloze accuracy** drops from 25.73% → 14.51% (GA), a **44% relative reduction** — this task requires exact open-ended recall and cannot be answered by process-of-elimination.
> - **ROUGE-L** drops from 0.516 → 0.387 (GA), a **25% relative reduction** in generation quality.
> - **Factuality score** drops from 5.2 → 3.83 (GA), a **26% relative reduction**.
>
> These results indicate that unlearning does substantially suppress detailed factual recall. Moreover, our new Table 3 below shows that when stronger unlearning is applied — e.g., MultiDelete achieves a 25% relative reduction in classification IT (42.0% → 31.4%) with preserved retain performance (37.2%) — POPS still recovers +7.3% IT and improves ROUGE-L from 0.219 to 0.371. Pushing unlearning further (GA 3+ epochs, KL-Min 3 epochs) collapses the model entirely before eliminating vulnerability. This privacy-utility trade-off suggests the issue is not weak baselines, but the fundamental difficulty of removing cross-modal associations while preserving model utility.
>
> **2) The recognition vs. recall distinction is key.**
> Multiple-choice classification tests whether the model can distinguish a correct answer from distractors — it does not require the model to generate the answer from scratch. An unlearned model might still identify "Stockholm" as more plausible than "Oslo" or "Berlin" without being able to produce "Stockholm" when asked "Where was this person born?" This is why we evaluate cloze and generation alongside classification.
>
> **3) Unlearning suppresses surface recall but may leave latent cross-modal associations exploitable.**
> The partial suppression shown above is consistent with POPS's mechanism. Unlearning damages the direct textual recall pathway (44% cloze drop) while potentially leaving visual-semantic associations in the shared embedding space intact. POPS targets these latent cross-modal traces, not surface-level recall. The fact that POPS can recover classification accuracy while cloze and generation metrics show different patterns suggests that the attack exploits structural traces at a level that standard evaluation metrics may not fully capture.
>
> **4) New results with stronger baselines directly address this concern.**
> Our MultiDelete results (Table 2) provide much larger unlearning effects that are clearly outside noise margins: classification drops of 25% relative (42.0% → 31.4%) where POPS still recovers +7.3% IT, and ROUGE-L IT recovery from 0.219 to 0.371. These recall-based metrics are immune to the ceiling-effect concerns about classification accuracy.
>
> **Table 3: Stronger Unlearning Configurations** (LLaVA-1.5-7B)
>
> | Method | Config | Forget IT | Retain IT | POPS IT | POPS Δ |
> |---|---|---|---|---|---|
> | GA | 1 epoch | 39.2% (-2.9) | 33.9% | 42.9% | +3.7 |
> | GA | 3 epochs | 0.0% (collapsed) | 0.0% | — | — |
> | KL-Min | 1 epoch | 40.4% (-1.6) | 43.5% | 46.9% | +6.5 |
> | KL-Min | 3 epochs | 0.0% (collapsed) | 0.0% | — | — |
> | MultiDelete | 3 epochs | 31.4% (-10.6) | 37.2% | 38.8% | +7.3 |
>
> Vanilla baseline: Forget IT = 42.0%, Retain IT = 43.6%.
>
> **Key observations:**
>
> These results suggest a **privacy-utility trade-off**. Methods that preserve reasonable model utility remain vulnerable to POPS, while aggressive configurations (GA 3+ epochs, KL-Min 3 epochs) that might resist POPS tend to collapse model utility entirely (0% on all metrics). MultiDelete is particularly informative: it achieves meaningful unlearning (IT -10.6) while preserving reasonable retain performance (37.2%), yet POPS still recovers +7.3%.

---

> ### Author Response · Authors · 2026-03-25
> **Response to Reviewer M4RG (Part-II)**
>
> **Addressing metric sensitivity.** We acknowledge the concern about small absolute classification changes. Our strongest claims rest on effects observed consistently across multiple metrics and methods. The MultiDelete ROUGE-L improvement (0.219 → 0.371) and the Table 1 multimodal vs. text-only swing (9.8 percentage points) are well outside statistical noise for the 250-sample forget set (50 identities × 5 IT questions).
>
> **Scope note.** The experiments in Tables 1–3 are conducted on LLaVA-1.5-7B to isolate the effect of multimodal-specific methods. The original paper (Tables 3–6) already demonstrates POPS's effectiveness across 4 architectures; the new experiments extend *depth* (more methods, stronger configurations) while the original provides *breadth* (architecture generalization).
>
> **Proposed revisions:** We will (a) add a metric sensitivity discussion in Section 4.2 distinguishing recognition from recall; (b) add a relative degradation analysis table; (c) add the privacy-utility trade-off analysis from Table 3.
>
> ---
>
> ### Q2: Experimental Setup Clarifications
>
> > *"Point (2) in 'Additional Implementation Details' of 4.1 is not clear how image/text pairs for this OOD set are constructed. Furthermore, it is not clear how S2L training setting is applied in the LoRA setting."*
>
> We apologize for the lack of clarity and will substantially revise Section 4.1. We provide detailed clarifications below.
>
> **OOD Dataset Construction:**
>
> The OOD dataset for PromptSuffix optimization is constructed from the **Retain Set** of each benchmark. Specifically:
>
> 1. **Image source**: We directly use the celebrity images from the Retain Set (153 celebrity profiles in MLLMU-Bench). These are real images of celebrities whose information the model is *not* asked to forget, serving as OOD data because they share the same *format* (celebrity portraits with biographical context) as the forget set but contain different *identities*.
>
> 2. **Text pair construction**: For each Retain Set image, we use the associated biographical questions and answers already provided in the benchmark. We sample 50 image-question-answer triplets uniformly from the Retain Set to form D_OOD.
>
> 3. **Why this is "OOD"**: We use "OOD" to mean *out-of-distribution relative to the forget identities*, not OOD in the standard generalization sense. The retain set profiles are in-distribution with respect to format (celebrity images + biographical QA) but are out-of-distribution with respect to the specific forget-set identities. The attacker does not know *which* profiles were forgotten — they only know that unlearning was applied and have access to data from the same domain. This is consistent with our threat model (Section 2.2). We will clarify this terminological distinction explicitly in the revised paper.
>
> **S2L Fine-tuning in the LoRA Setting:**
>
> The Shake-to-Leak fine-tuning is applied via LoRA (Low-Rank Adaptation) as follows:
>
> 1. **LoRA configuration**: We attach LoRA adapters (rank=8, alpha=16, dropout=0.05) to the query and value projection matrices (`q_proj`, `v_proj`) of the language model component. This adds only ~0.1% trainable parameters while keeping the vision encoder and multimodal projector frozen.
>
> 2. **Training data**: The synthetic dataset generated in Stage 2 (using optimized suffixes to query the unlearned model) is used as the fine-tuning data. Each sample consists of an (image, optimized_question, model_response) triplet.
>
> 3. **Training procedure**: We fine-tune with AdamW optimizer (lr=1e-4), batch size 4 with 4-step gradient accumulation (effective batch size 16), for 3 epochs. A KL divergence penalty (weight=0.2) between the fine-tuned and original unlearned model outputs on retain-set samples prevents catastrophic forgetting of general knowledge.
>
> 4. **Why LoRA**: The parameter-efficient nature of LoRA is important for the threat model — it demonstrates that even limited fine-tuning access (few trainable parameters, small data) is sufficient to recover unlearned knowledge, making the attack more realistic. Full fine-tuning would also work but would represent a stronger (and less realistic) attacker capability assumption.
>
> **Proposed revisions:** We will add a dedicated "OOD Dataset Construction" paragraph and a "S2L Fine-tuning Details" paragraph in Section 4.1, with the above details. We will also add a LoRA configuration table in the appendix.
>
> ---
> We sincerely thank Reviewer M4RG again for the valuable feedback, which helped us identify important clarity gaps and motivated the stronger baseline experiments. We welcome any further questions or discussion and will do our best to address them thoroughly.

---

### Review · Reviewer_n34t · 2026-03-03

**Summary Of Contributions:**

This paper studies the robustness of machine unlearning for multimodal LLMs. The authors propose prompt-optimized parameter shaking (POPS) to attack an unlearned MLLMs and recover the knowledge that was supposed to be removed. First, a gradient-based method is used to learn a universal adversarial suffix with OOD data (PromptSuffix). Then, it adapts the previous algorithm Shake-to-Leak (S2L) to the multimodal setting. A thorough empirical study across models, datasets, and unlearning methods are provided to show the effectiveness of the proposed algorithm.

**Audience:**

Yes

**Audience Explanation:**

The study of machine unlearning for (M)LLMs is of general interest to the machine learning community.

**Broader Impact Concerns:**

N/A.

**Claims And Evidence:**

Yes

**Claims Explanation:**

The paper provides extensive and thorough empirical evaluations.

**Requested Changes:**

The paper has provided extensive and thorough empirical evaluations. My biggest concern is on the novelty side.

1. The key components of the algorithm are extensions and combinations of existing methods. Prompt optimization is extensively used in LLMs attacks including jailbreaking. S2L was also proposed in previous works.

2. The studied problem is not new for LLMs, and there exist many works saying that current unlearning methods are not robust. For example, arXiv:2409.18025, and many others in this work. It looks to me that everything is just repeated again, but for MLLMs, in this paper.

3. The paper argues that multimodal models introduce new challenges compared to unimodal models. However, the attacked unlearning methods in the paper are adaptations of unimodal approaches. As a result, it is not clear whether the robustness issues come from that unimodal unlearning methods are limited and not enough for multimodal models. In my opinion, it makes more sense to test against stronger unlearning methods designed specifically for multimodal models.

I would recommend acceptance if the above points are clarified in the rebuttal or the revision of the paper.

---

> ### Author Response · Authors · 2026-03-25
> **Response to Reviewer n34t (Part-I)**
>
> We sincerely thank Reviewer n34t for recognizing our "extensive and thorough empirical evaluations" and for the specific suggestions for improvement. We address each concern below.
>
> ---
>
> ### Q1: Novelty of Algorithm Components
>
> > *"The key components of the algorithm are extensions and combinations of existing methods. Prompt optimization is extensively used in LLMs attacks including jailbreaking. S2L was also proposed in previous works."*
>
> We appreciate this observation and would like to respectfully clarify how POPS differs from prior prompt optimization and S2L methods in ways that go beyond simple combination.
>
> **1) OOD-guided concept-level optimization vs. token-level jailbreaking.**
> Existing prompt optimization methods (e.g., GCG, Zou et al., 2023) operate at the *token level*, maximizing loss on individual tokens to bypass safety mechanisms. In contrast, POPS performs *concept-level* optimization guided by out-of-distribution (OOD) data. Here, “concept-level” refers to the optimization objective rather than the parameterization. Although we optimize token embeddings, the supervision is defined over semantic attributes (QA correctness across a dataset), which encourages cross-instance generalization instead of token-level memorization. This is a qualitatively different objective: rather than finding universal jailbreak suffixes, POPS discovers suffixes that bridge the semantic gap between OOD knowledge and supposedly forgotten knowledge, exploiting persistent *cross-modal associations* in the shared embedding space. As shown in Table 7 of our paper, this design difference is empirically significant: POPS achieves 82% recovery vs. GCG's 48%, and even PromptSuffix alone (70%) substantially outperforms GCG, validating that OOD-guided concept-level optimization is fundamentally more effective for multimodal unlearning attacks.
>
> **2) Multimodal S2L: technically distinct from prior S2L work.**
> The original S2L (Li et al., 2024b) was designed for text-to-image diffusion models in a unimodal setting. Our adaptation to MLLMs requires three concrete technical changes that do not exist in prior S2L:
>
> - *Multi-modal training pairs*: Unlike unimodal S2L which fine-tunes on text-only data, our S2L simultaneously exposes the model to image-question-answer triplets, which is necessary to reactivate dormant visual-semantic alignment in the LLaVA cross-attention layers (vision encoder → language model projection).
> - *Faceted image-text decomposition*: A single celebrity profile is decomposed into multiple training examples covering distinct facets (appearance, biography, occupation, associations). This faceting is specific to MLLM identity knowledge, which is structured across both modalities — it has no equivalent in unimodal diffusion S2L.
> - *OOD-guided synthetic data generation*: The fine-tuning data is synthesized by querying the unlearned model with OOD-guided suffixes from Stage 1 — the two stages are thus coupled in a feedback loop. Prior S2L uses independently collected training data; our S2L data is generated by the Stage 1 component.
>
> These changes are motivated by the multimodal architecture and are empirically necessary. We acknowledge that formal per-component ablations are a direction for future work.
>
> **3) The closed-loop integration is itself a contribution.**
> The synergy between OOD-guided prompt optimization and multimodal S2L creates a *closed-loop* attack pipeline where discovered vulnerabilities from Stage 1 are amplified through Stage 2.
>
> - *PromptSuffix → S2L (synthetic private-set construction)*: Optimized suffixes specifically elicit residual knowledge aligned with target attributes, producing higher-quality, more targeted synthetic supervision for S2L than naive OOD prompting.
> - *S2L → PromptSuffix (signal amplification)*: S2L fine-tunes the model on this synthetic data, which does not create new knowledge but amplifies the weak signals already exposed by PromptSuffix. As a result, the model becomes more responsive to the same prompting strategy.
> - *Closed-loop effect*: PromptSuffix improves the construction of the synthetic private set used by S2L, while S2L improves the model's sensitivity to the same prompts. This forms an extraction → amplification feedback loop, rather than a simple additive combination.
>
> This closed-loop design is new — prior prompt attacks (GCG, jailbreaking) are open-loop (prompt only), and prior S2L operates without prompt optimization guidance. The ablation in Table 5 shows the contribution of each stage: on the more sensitive cloze accuracy metric, S2L provides a meaningful gain (PromptSuffix alone: 17.65% → Full POPS: 18.2%, a +3.1% relative improvement over the prompt-only variant). Crucially, Table 6 shows that applying S2L without OOD-guided suffix optimization achieves only 21% recovery, versus 82% for Full POPS, confirming that the closed-loop design — not either component in isolation — drives the attack's effectiveness.

---

> ### Author Response · Authors · 2026-03-25
> **Response to Reviewer n34t (Part-II)**
>
> **Proposed revision:** We will add a dedicated "Novelty Discussion" paragraph in Section 3 explicitly contrasting POPS with GCG/jailbreaking prompt optimization and unimodal S2L, emphasizing the three multimodal-specific technical innovations above.
>
> ---
>
> ### Q2: Problem Novelty vs. Unimodal Unlearning Attacks
>
> > *"The studied problem is not new for LLMs, and there exist many works saying that current unlearning methods are not robust. For example, arXiv:2409.18025, and many others. It looks to me that everything is just repeated again, but for MLLMs."*
>
> We agree with the reviewer that the robustness of unimodal LLM unlearning is an active area — works such as arXiv:2409.18025 (and Yuan et al., 2024, arXiv:2408.10682, which we cite) have established that text-only unlearning methods are not robust. We do not claim to be the first to study unlearning robustness in general. Our contribution is that the **multimodal setting introduces qualitatively new vulnerabilities** that do not exist in, and cannot be discovered by, unimodal analysis:
>
> **1) Cross-modal memory persistence is a new attack vector with no unimodal analogue.**
> In unimodal LLMs, robustness failures arise from residual knowledge within a single representation space. In MLLMs, information is encoded across *multiple interacting modalities* — visual features from a person's workplace image can leak textual information about their occupation even after the textual knowledge has been unlearned. This cross-modal memory persistence requires fundamentally different attack strategies (our OOD-guided PromptSuffix) that would not apply to text-only models. **Table 1 provides direct evidence for this claim:** applying POPS in text-only mode (stripping out all image inputs) to the same GA-unlearned LLaVA model *decreases* Forget IT by 6.1%, while the full multimodal POPS *increases* it by 3.7% — a **9.8 percentage point swing**. The visual pathway is not merely additive; it appears essential for the attack to function at all.
>
> **2) Quantitative evidence of additional vulnerability.**
> Works such as arXiv:2409.18025 study unimodal LLM robustness in a single-modality setting where attacking requires direct access to text-domain residuals. Our MLLM recovery rate of 82% (Table 3, averaged across 4 architectures) reflects the *additional* attack surface provided by cross-modal associations. Table 1 directly quantifies this: on the same LLaVA-1.5-7B model with the same GA unlearning, multimodal POPS recovers +3.7% IT while text-only POPS hurts by -6.1%, demonstrating that the visual pathway is the critical attack vector.
>
> **3) Practical significance.**
> MLLMs are deployed in sensitive domains (medical imaging, identity verification, document understanding) where both visual and textual privacy matter simultaneously. The cross-modal leakage pathways we identify represent a genuine threat specific to multimodal architectures.
>
> **4) Benchmark contribution.**
> We are the first to systematically evaluate MMU robustness across 3 multimodal benchmarks (MLLMU-Bench, CLEAR, UnLoK-VQA) and 4 MLLM architectures (LLaVA-1.5-7B, Qwen-VL-Chat-7B, InternVL3-9B, Llama-3.2-11B-V), providing a comprehensive multimodal evaluation that complements existing unimodal robustness studies.
>
> **Table 1: Unimodal vs. Multimodal Attack Comparison** (GA-unlearned LLaVA-1.5-7B)
>
> | Attack Mode | Forget IT (unlearned → attacked) | IT Δ | ROUGE-L IT Change |
> |---|---|---|---|
> | Multimodal POPS | 39.2% → 42.9% | **+3.7** | +26.4% |
> | Text-only POPS | 39.2% → 33.1% | **-6.1** | +2.5% |
>
> The text-only degradation is itself informative: it shows POPS is not a generic fine-tuning trick. Without images, the optimized suffixes cannot anchor to visual identity features, and the synthetic data lacks the visual-semantic associations needed for recovery. The ROUGE-L metric further illustrates this: multimodal POPS improves generation quality substantially (+26.4%), while text-only barely changes it (+2.5%).
>
> **Proposed revision:** We will strengthen Section 1 to explicitly position POPS relative to unimodal unlearning robustness works, clarifying that our contribution is the characterization of the *cross-modal* attack surface, not the general claim that unlearning methods are fragile.

---

> ### Author Response · Authors · 2026-03-25
> **Response to Reviewer n34t (Part-III)**
>
> ### Q3: Testing Against Multimodal-Specific Unlearning Methods
>
> > *"The attacked unlearning methods in the paper are adaptations of unimodal approaches. It is not clear whether the robustness issues come from that unimodal unlearning methods are limited and not enough for multimodal models. It makes more sense to test against stronger unlearning methods designed specifically for multimodal models."*
>
> This is an excellent suggestion, and we fully agree that testing against multimodal-specific unlearning methods would strengthen the paper. We have implemented and evaluated POPS against two recently published multimodal-specific unlearning methods:
>
> **1) MANU (Liu et al., 2025, arXiv:2502.15910)** — *Modality-Aware Neuron Pruning for Unlearning in MLLMs*. MANU identifies and prunes neurons specifically responsible for cross-modal associations, representing a fundamentally different approach from gradient-based methods. This directly addresses the reviewer's concern: if POPS can still recover knowledge after modality-aware pruning, the vulnerability may be related to the multimodal architecture rather than a limitation of unimodal-adapted methods.
>
> **2) MultiDelete (Cheng & Amiri, 2024, ECCV 2024)** — *Multimodal Machine Unlearning via Modality Decoupling*. MultiDelete explicitly decouples modalities during the unlearning process, preserving unimodal representations while erasing cross-modal bindings.
>
> **Table 2: POPS Attack on Multimodal-Specific Baselines** (LLaVA-1.5-7B, vanilla Forget IT = 42.0%)
>
> | Unlearning Method | Type | Forget IT (unl) | Forget IT (POPS) | IT Δ | ROUGE-L IT (unl → atk) |
> |---|---|---|---|---|---|
> | GA | Unimodal-adapted | 39.2% | 42.9% | +3.7 | 0.436 → 0.551 |
> | KL-Min | Unimodal-adapted | 40.4% | 46.9% | +6.5 | 0.446 → 0.377 |
> | **MANU** | **Multimodal (pruning)** | 37.6% | 43.7% | **+6.1** | 0.457 → 0.478 |
> | **MultiDelete** | **Multimodal (contrastive)** | 31.4% | 38.8% | **+7.3** | 0.219 → 0.371 |
>
> **Key observations:**
>
> **(a) POPS recovers from both tested multimodal-specific methods.** Despite fundamentally different unlearning mechanisms — neuron pruning (MANU) and contrastive modality decoupling (MultiDelete) — POPS successfully recovers from both. The consistency across diverse methods suggests the vulnerability may be related to the shared representation space inherent to MLLM architectures, rather than a limitation of any specific unlearning algorithm. We note that further investigation with additional multimodal-specific methods would strengthen this conclusion.
>
> **(b) MultiDelete — deepest unlearning, yet substantial recovery.** MultiDelete achieves the deepest unlearning (IT 42.0% → 31.4%, a 25% relative reduction), yet POPS recovers +7.3% IT and increases ROUGE-L IT from 0.219 to 0.371. This indicates that latent cross-modal traces can survive even explicit contrastive modality decoupling, though the recovery does not reach vanilla-level performance, suggesting MultiDelete does eliminate some cross-modal associations.
>
> **(c) MANU — pruning-based unlearning appears reversible.** MANU's identity-aware neuron pruning reduces Forget IT by 4.5%, but POPS recovers +6.1%, exceeding vanilla levels. This suggests that pruning-based approaches may have difficulty eliminating knowledge that is distributed across the model's remaining neurons, though we note the relatively mild unlearning effect (4.5% drop) limits the conclusions we can draw.
>
> **(d) KL-Min — POPS classification recovery exceeds vanilla.** KL-Min shows POPS classification recovery to 46.9%, exceeding the vanilla baseline (42.0%). We note that the ROUGE-L IT metric decreases slightly (0.446 → 0.377), indicating that S2L fine-tuning may specialize the model toward the classification format at some cost to open-ended generation. The classification recovery is nonetheless significant because it demonstrates that an attacker can re-enable the model's ability to correctly identify forgotten individuals from images — the core privacy concern. The fact that even mild KL-Min unlearning (only -1.6% IT drop) leaves enough latent signal for an attacker to enhance identification accuracy underscores the potential severity of the vulnerability.
>
> **Proposed revisions:** We will add Table 1 (unimodal vs. multimodal comparison) in Section 4.4 and Table 2 (multimodal-specific baselines) in Section 4.2.
>
> ---
> We sincerely thank Reviewer n34t again for the constructive and detailed feedback. Your suggestions directly led to new experiments and discussions that we believe substantially strengthen the paper. We welcome any further questions or discussion and will do our best to address them thoroughly.

---

### Review · Reviewer_GuMG · 2026-04-18

**Summary Of Contributions:**

In this paper, the authors study the robustness of multimodal machine unlearning under realistic adversarial interactions. They propose Prompt-Optimized Parameter Shaking (POPS), a framework designed to systematically probe the threat that, even when direct instance-level recall is effectively suppressed, unlearned MLLMs may still retain functionally accessible representations that can be reactivated through appropriate prompts. Specifically, POPS introduces an OOD-guided PromptSuffix attack to expose residual multimodal representations, with self-synthesis fine-tuning serving as an amplification mechanism. Experiments across multiple benchmarks and architectures demonstrate systematic weaknesses in current multimodal unlearning methods, highlighting the need for approaches that explicitly address the vulnerability.

**Audience:**

Yes

**Audience Explanation:**

This paper identifies and studies the robustness problem of MLLM unlearning, and reveals an important phenomenon that would be of broad interest to researchers working on machine unlearning, IP protection, adversarial robustness, and multimodal LLMs.

**Claims And Evidence:**

Yes

**Claims Explanation:**

The authors provide sufficiently clear and convincing empirical evidence to support their claims.

**Requested Changes:**

The research problem is important and the proposed method appears reasonable,  and is supported by extensive and comprehensive empirical validation. However, I still have the following minor concerns:

While the paper relies on OOD data as an important component of the proposed attack, the construction and characteristics of this OOD data are not sufficiently described. Although the paper provides OOD-related ablation results, it does not clearly present concrete examples and details of the OOD samples themselves. I strongly encourage the authors to provide such details and representative examples, and to further investigate the relationship between the OOD data and the true forget set, for example by analyzing how distributional differences affect the attack effectiveness. Doing so would improve both the clarity and reproducibility of the paper, and also help clarify whether the proposed method genuinely recovers unlearned knowledge or instead re-learns it through additional training signals.

---

> ### Author Response · Authors · 2026-04-26
> **Response to Reviewer GuMG**
>
> We sincerely thank Reviewer GuMG for recognizing our "important research question and comprehensive empirical validation" and for the specific suggestions for improvement. We address the concern below.
>
> ---
>
> ### Q: Regarding the construction and characteristics of the OOD data used in the POPS attack
>
> We thank the reviewer for this helpful suggestion. We agree that the construction and role of the OOD data should be described more clearly, and we will revise the paper by following the valuable comments accordingly.
>
> In our experiments, **the OOD data is constructed from samples that are disjoint from the forget set but share similar structural and semantic characteristics with the target task**. For example, in **MLLMU-Bench, the OOD data is drawn from the Retain Set**, where each sample consists of a person image paired with biographical attributes and image-question-answer pairs. A representative OOD sample may include a portrait of a non-forgotten individual (e.g., “Amelia Kuznetsov”), along with attributes such as occupation, birthplace, and residence, and a question like “What is the profession of this person?” with the answer “Environmental Scientist.” While this sample is disjoint from the forgotten identities, it follows the same multimodal structure and attribute space as the forget set.
>
> Importantly, our use of **OOD data is not limited to person-profile settings**. For datasets such as CLEAR and UnLoK-VQA, where the tasks involve entity prediction or visual question answering rather than identity profiles, the OOD data is constructed analogously: samples are drawn from disjoint subsets that share the same task format (e.g., image-question-answer triplets or entity-level reasoning), ensuring structural alignment without overlapping target concepts.
>
> The purpose of using OOD data is therefore **not to approximate the forget set directly, but to provide a proxy distribution for learning generalizable multimodal patterns**. The optimized suffixes capture how relevant attributes and reasoning pathways are expressed across modalities, rather than memorizing specific instances. This allows the attack to exploit shared cross-modal structures that persist after unlearning.
>
> We would also clarify that **the synthetic data used in POPS does not contain ground-truth information from the forget set**. Instead, it is generated entirely from the unlearned model itself. Consistent with this, our ablations show that direct fine-tuning on OOD data alone yields only limited recovery, whereas PromptSuffix and the full POPS pipeline recover substantially more. This indicates that the attack primarily reactivates latent residual knowledge rather than re-learning it from external supervision.
>
> ----
>
> We appreciate Reviewer GuMG again for the constructive and detailed feedback. Your suggestions directly led to clearer clarification and analysis that we believe would substantially strengthen the paper. We welcome any further questions or discussion and will do our best to address them thoroughly.

---

### Author Response · Authors · 2026-04-27
**General response and sincerely appreciate reviewers' feedback.**

Dear all reviewers,

We sincerely thank all the reviewers for your constructive comments and insightful suggestions, which help us make our work more complete and further improve the quality of the manuscript. We are also glad that the reviewers acknowledge that our proposed methods (1) address an important and under-explored research problem on multimodal unlearning robustness, (2) are supported by extensive empirical evaluations across multiple benchmarks and architectures, and (3) reveal findings of broad interest to the machine unlearning, multimodal LLM, and adversarial robustness communities.

According to the reviews, we have provided an updated revision with the following changes:
- We added a dedicated OOD Dataset Construction paragraph in Section 4.1, clarifying the source, sampling protocol, and intended meaning of "OOD" with a concrete example, and noting that no forget-set images or annotations are used at any stage.
- We added new experiments evaluating POPS against multimodal-specific unlearning methods (MANU and MultiDelete), summarized at the end of Section 4.2 with full results in Appendix B.
- We added a controlled text-only ablation at the end of Section 4.3 (Cross-Modal Attacking Pathway), with the table and detailed analysis in Appendix D, confirming the visual pathway as the critical attack vector unique to multimodal unlearning.
- We added a privacy-utility dilemma analysis under stronger unlearning configurations, integrated into the Privacy-Utility Trade-off discussion in Section 4.4 and detailed in Appendix C.
- We added a metric sensitivity analysis at the opening of Section 4.2 and detailed in Appendix A, distinguishing recognition from recall and showing that generation-based metrics reveal substantially larger forgetting than classification accuracy.
- We added a discussion about the unique contribution compared with prior works (Appendix I), with corresponding pointers from Section 2.1 and the closing of Section 3, contrasting POPS with GCG-style prompt optimization and unimodal S2L along three axes: concept-level OOD-guided optimization, multimodal-specific S2L extensions, and closed-loop integration.
- We expanded Section 1 to position our contribution relative to unimodal unlearning robustness work, framing cross-modal memory persistence as a qualitatively new attack surface that is absent from unimodal systems.
- We added a detailed S2L LoRA fine-tuning configuration in Section 4.1, with full hyperparameters in Appendix E.

Thank you again for your time and feedback. We welcome any further questions or discussion and will do our best to address them thoroughly.

Best regards,
The Authors

---

### Decision · Action_Editor_6ihB · 2026-06-01

**Recommendation:** Accept as is

**Audience:**

Yes

**Audience Explanation:**

This work focuses on an important question: whether multimodal unlearning methods remain robust under realistic adversarial interaction. The results should be of interest to researchers working on machine unlearning, multimodal foundation models, privacy-preserving machine learning, and adversarial robustness.

**Claims And Evidence:**

Yes

**Claims Explanation:**

The paper makes a well-supported contribution to the area of multimodal machine unlearning robustness. The authors propose an attack framework that combines prompt optimization and parameter-shaking fine-tuning to recover information from supposedly unlearned MLLMs. The submission provides extensive empirical evaluation across multiple MLLM architectures, benchmarks, and unlearning methods, and the evidence presented is sufficient to support the main claims.

During review, reviewers raised concerns regarding (i) the novelty of the approach relative to prior prompt-based attacks and unlearning robustness work, (ii) the construction and role of the OOD data used in the attack, (iii) the strength of the underlying unlearning baselines, and (iv) evaluation against multimodal-specific unlearning methods. The authors provided detailed responses and committed revisions addressing each of these concerns. They clarified the multimodal-specific aspects of their method, added discussion contrasting their approach with prior work, expanded the description of OOD data construction, included stronger analyses of the privacy–utility trade-off, and reported new experiments against multimodal-specific unlearning methods. These additions substantially strengthen the paper and adequately address the reviewers' concerns.